# New footprints from Laetoli (Tanzania) provide evidence for marked body size variation in early hominins

**Fidelis T Masao[1], Elgidius B Ichumbaki[1], Marco Cherin[2,3]\*, Angelo Barili[4], Giovanni Boschian[5], Dawid A Iurino[3,6], Sofia Menconero[7], Jacopo Moggi-Cecchi[8], Giorgio Manzi[9]**

[1]Department of Archaeology and Heritage Studies, University of Dar es Salaam, Dar es Salaam, Tanzania; [2]Dipartimento di Fisica e Geologia, Università di Perugia, Perugia, Italy; [3]PaleoFactory, Sapienza Università di Roma, Roma, Italy; [4]Galleria di Storia Naturale, Centro d'Ateneo per i Musei Scientifici, Università di Perugia, Perugia, Italy; [5]Dipartimento di Biologia, Università di Pisa, Pisa, Italy; [6]Dipartimento di Scienze della Terra, Sapienza Università di Roma, Roma, Italy; [7]Studio Associato Grassi, Perugia, Italy; [8]Dipartimento di Biologia, Università di Firenze, Firenze, Italy; [9]Dipartimento di Biologia Ambientale, Sapienza Università di Roma, Roma, Italy

**Abstract** Laetoli is a well-known palaeontological locality in northern Tanzania whose outstanding record includes the earliest hominin footprints in the world (3.66 million years old), discovered in 1978 at Site G and attributed to *Australopithecus afarensis*. Here, we report hominin tracks unearthed in the new Site S at Laetoli and referred to two bipedal individuals (S1 and S2) moving on the same palaeosurface and in the same direction as the three hominins documented at Site G. The stature estimates for S1 greatly exceed those previously reconstructed for *Au. afarensis* from both skeletal material and footprint data. In combination with a comparative reappraisal of the Site G footprints, the evidence collected here embodies very important additions to the Pliocene record of hominin behaviour and morphology. Our results are consistent with considerable body size variation and, probably, degree of sexual dimorphism within a single species of bipedal hominins as early as 3.66 million years ago.

\*For correspondence: marco.cherin@unipg.it

**Competing interests:** The authors declare that no competing interests exist.

## Introduction

Estimates of body size and proportions are crucial in the evolutionary interpretation of Plio-Pleistocene hominin palaeobiology (*McHenry, 1991*, *1992*; *Ruff et al., 1997*; *Grabowski et al., 2015*) and have been the subject of ongoing debates, at least since the late 1970s (e.g., *Johanson and White, 1979*). Within-species variability in body size often relates to sexual dimorphism and/or to adaptation to different ecologies. This is particularly true among extant Hominoidea, which show diverse patterns of variation (e.g., *Plavcan, 2001*); for instance, gorillas are polygynous species with strong sexual dimorphism due to intense male-male competition, whereas chimpanzees are promiscuous with definitively smaller sexual dimorphism. It is reasonable to assume that complex relationships among body size, sexual dimorphism, mating system (and/or reproductive strategy) and social structure/behaviour also applied to extinct hominins, including our bipedal relatives of the Plio-Pleistocene. In fact, claims that size variation in *Australopithecus* and/or *Paranthropus* was larger than that in recent human populations include inferences on sexual dimorphism (*Richmond and Jungers, 1995*; *Plavcan et al., 2005*; *Lockwood et al., 2007*; but see *Reno et al., 2003*), whereas arguments

**eLife digest** Fossil footprints are extremely useful tools in the palaeontological record. Their physical features can help to identify their makers, but can also be used to infer biological information. *How did the track-maker move? How large was it? How fast was it going?*

Footprints of hominins (namely the group to which humans and our ancestors belong) are pretty rare. Nearly all of the hominin footprints discovered so far are attributed to species of the genus *Homo*, to which modern humans belong. The only exceptions are the footprints that were discovered in the 1970s at Laetoli (in Tanzania) on a cemented ash layer produced by a volcanic eruption. These are thought to have been made by three members of the hominin species *Australopithecus afarensis* – the same species as the famous "Lucy" from Ethiopia – around 3.66 million years ago.

The extent to which body shape and size varied between different members of *Au. afarensis* – for example, between males and females – has been the subject of a long debate among researchers. Based on the skeletal remains found so far in East Africa, some scholars believe that individuals only varied moderately, as in modern humans, while others state that it was pronounced, as in some modern apes like gorillas.

Masao et al. have now unearthed new bipedal footprints from two individuals who were moving on the same surface and in the same direction as the three individuals who made the footprints documented in the 1970s. The estimated height of one of the new individuals (about 1.65 metres) greatly exceeds those previously published for *Au. afarensis*. This evidence supports the theory that body size varied considerably amongst individuals within the species.

Masao et al. tentatively suggest that the new footprints can be considered as a whole with the 1970s ones. The tall individual may have been the dominant male of a larger group, the others smaller females and juveniles. Thus, considerable differences may have existed between males and females in these remote human ancestors, similar to modern gorillas.

The newly discovered tracks are only 150 metres away from the previously discovered sets of footprints. This leaves open the possibility that additional tracks may be unearthed nearby that will further our knowledge about the variability and behaviour of our extinct ancestors.

referring to early *Homo* are usually associated with eco-physiological variants (*Antón et al., 2014*; *Di Vincenzo et al., 2015*).

For *Australopithecus afarensis*, remarkable variation in size and shape within its alleged hypodigm was noted in the original description of the species (*Johanson et al., 1978*). Nevertheless, there have always been disputes about the nature and degree of sexual dimorphism characterising this early bipedal hominin, with supporters of either pronounced (e.g., *Johanson and White, 1979*; *Kimbel and White, 1988*; *McHenry, 1991*; *Richmond and Jungers, 1995*; *Lockwood et al., 1996*; *Plavcan et al., 2005*; *Harmon, 2006*; *Gordon et al., 2008*) or moderate (*Lovejoy et al., 1989*) body-size dimorphism.

For example, *Richmond and Jungers (1995)* wrote: 'If the fossils from Hadar and Maka (and Laetoli) are assumed [...] to be from one sexually dimorphic species, then the degree of sexual dimorphism of *Au. afarensis* would have been at least as extreme as that of the most dimorphic living apes [...]. It follows that a strictly monogamous structure would have been highly unlikely.' *Reno et al. (2003)* (but see *Plavcan et al., [2005]* and the reply by *Reno et al., [2005]*) challenged this premise with an analysis of the sexual dimorphism of femoral head diameter in *Au. afarensis*, concluding that these early hominins showed human-like sexual dimorphism and were therefore characterised by a monogamous mating system. Conversely, *Grabowski et al. (2015*, p. 90) obtained comprehensive and thoroughly vetted data, supporting 'arguments that *Au. afarensis* had substantial size dimorphism [...] leading to a large amount of variation in body size within this taxon.'

It is clear that our ability to investigate this important and controversial issue depends on the possibility of evaluating the body size and proportions of extinct creatures. Estimates are largely inferred from known relationships between metric data in living species, such as bone length (or joint size)

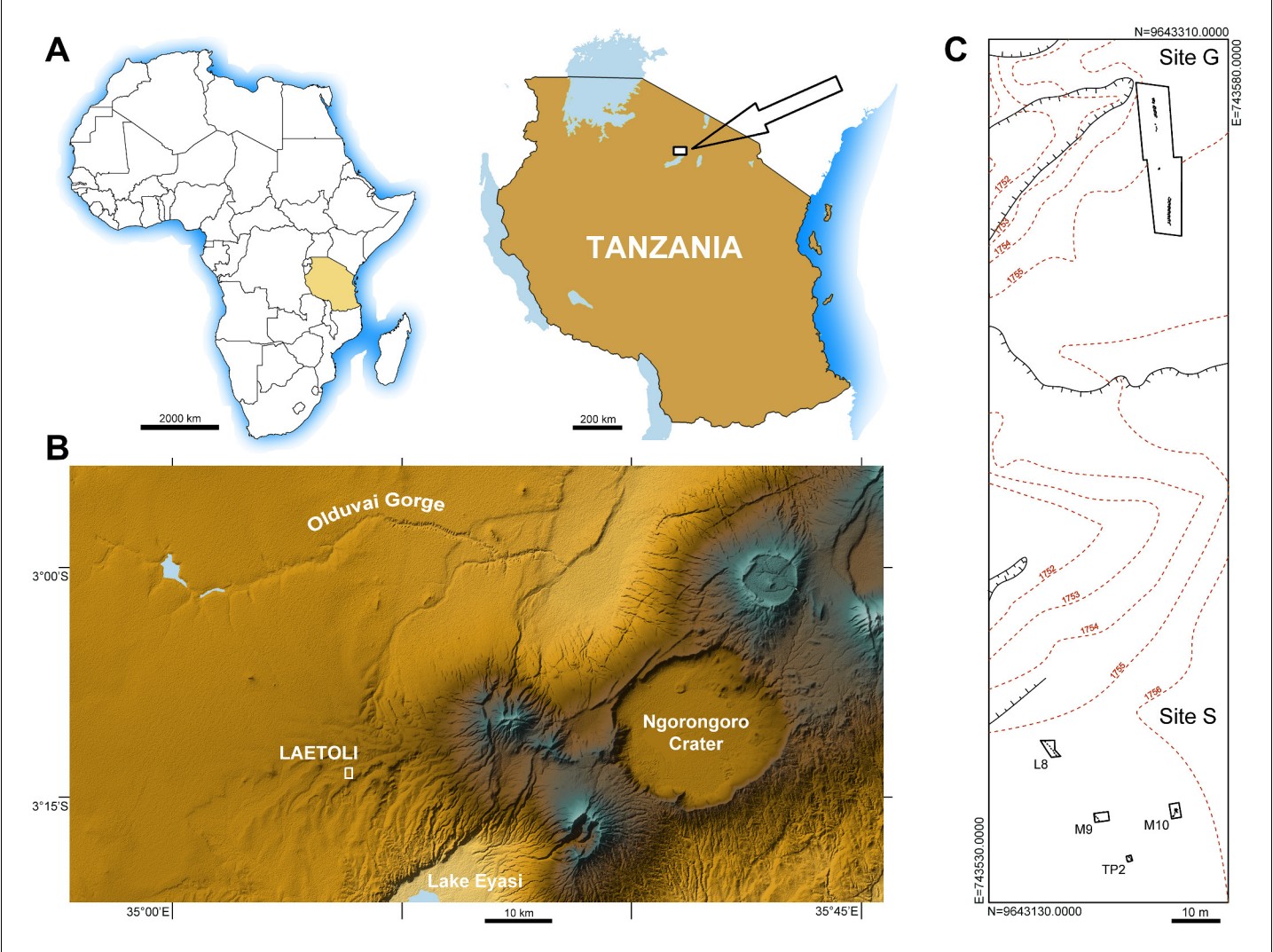

**Figure 1.** Geographical location and site map. (**A**) Location of the study area in northern Tanzania. (**B**) Location of Laetoli within the Ngorongoro Conservation Area, about 50 km south of Olduvai Gorge. (**C**) Plan view of the area of Laetoli Locality 8 (Sites G and S).

and stature (or body mass) (*McHenry, 1991*, *1992*; *Grabowski et al., 2015*). Similar estimates can be even more plainly obtained from the analysis of single footprints or – even better – from trails of footprints (*Tuttle, 1987*; *Dingwall et al., 2013*). Among these, one of the most remarkable pieces of evidence are the renowned trackways from Laetoli Site G (northern Tanzania), which are ascribed to *Au. afarensis* (*White and Suwa, 1987*).

In this paper, we report a novel set of hominin tracks discovered at Laetoli in the new Site S, comparing it to a reappraisal of the original evidence. The new tracks can be referred to two different individuals moving in the same direction and on the same palaeosurface as those documented at Site G.

## The site: a brief overview

Laetoli (*Figure 1A,B*) is one of the most important palaeontological localities in Africa. It lies within the Ngorongoro Conservation Area at the southern edge of the Serengeti Plains. The region includes sites such as Olduvai Gorge, Lake Ndutu and Laetoli itself and provides a long sequence of Plio-Pleistocene, mostly volcano-sedimentary, deposits that are rich in archaeological and paleonto-logical remains (*Hay, 1987*), overlying Precambrian metamorphic rocks. The paleoanthropological

significance of the whole area has been known since the mid 1930s (*Reck and Kohl-Larsen, 1936*; *Kohl-Larsen, 1943*), whereas Laetoli became known worldwide in the 1970s for stimulating discoveries, such as the holotype and other remains of *Au. afarensis* (*Leakey et al., 1976*; *Johanson et al., 1978*) and remarkable evidence of the earliest bipedal hominin tracks (*Leakey and Hay, 1979*; *Leakey and Harris, 1987*) dated to 3.66 million years ago (Ma) (*Deino, 2011*).

Mammal, bird and insect prints and trails have been identified in 18 sites (labelled from A to R) out of 33 total palaeontological localities in the Laetoli area (*Leakey, 1987a*; *Musiba et al., 2008*; *Harrison and Kweka, 2011*). Footprints occur in 10 sublevels within the so-called Footprint Tuff, corresponding to the lower part of Tuff 7 in the Upper Laetolil Beds stratigraphic sequence (*Hay, 1987*). These hominin trackways were found in 1978 at Site G (Locality 8) and were referred to three individuals (G1, G2, G3) of different body size: the smallest individual, G1, walked side by side on the left of the largest individual, G2, while the intermediate-sized individual, G3, superimposed its feet over those of G2 (*Leakey, 1981*). The trackways are usually ascribed, not without controversy (*Tuttle et al., 1991*; *Harcourt-Smith, 2005*), to *Au. afarensis* (*White and Suwa, 1987*), which is the only hominin taxon found to date in the Upper Laetoli Beds (*Harrison, 2011*).

## Discovery and notes on preservation

The new Site S (situated within Locality 8) is located about 150 m to the south of Site G (*Figure 1C*), on the surface of the same morphological terrace. It was discovered during systematic survey and excavation activities (Cultural Heritage Impact Assessment) aimed at evaluating the impact of a proposed new field museum at Laetoli, in the area of Locality 8. Sixty-two 2 × 2 m test pits were randomly positioned within a grid and were carefully excavated down to the Footprint Tuff and sometimes deeper.

In 2015, fourteen hominin tracks always associated with tracks of other vertebrates (see Results) were unearthed in three test-pits, respectively labelled L8, M9 and TP2 from north to south (see Materials and methods) (*Figures 1C* and *2*). Seven bipedal tracks in different preservation state (see below) were exposed in L8 (*Figure 2*; *Figure 2—figure supplement 1* and *Figures 3–4*) and four in M9 (*Figure 2—figure supplement 2* and *Figure 5*). Two additional tracks of the same individual were found in the eastern part of TP2 (*Figure 6*). All these prints are clearly referable to a single individual trackway, with an estimated total length of 32 m and trending SSE to NNW (i.e., 320–330°), approximately parallel to the G1 and G2/3 trackways. Following the code used for the Site G prints (*Leakey, 1981*), we refer to the new individual as S1 (footprint numbers S1-1–7 in L8, S1-1–4 in M9 and S1-1–2 in TP2). At the end of the September 2015 field season, we discovered one more track referable to a second individual (S2), in the SW corner of TP2. Conversely, we exposed only non-hominin footprints in test-pit M10 (*Figure 2—figure supplement 3*).

The preservation state of the tracks varies considerably along the trackway, depending on the depth of the Footprint Tuff from the surface.

In L8, the Tuff is very shallow, not deeper than 20 cm to the south, whereas it even crops out on the scarp of the terrace on the opposite side. Consequently, the Tuff is overlain here only by reworked loose soil, and the tracks are not filled up with compact and/or cemented sediment. Preservation issues arise from this situation, because the tuff tends to be rather altered and dislodged along the natural fractures (*Figure 7*). The first four tracks in the L8 trail are the best preserved, whereas the state of preservation of the footprint-bearing surface is particularly critical in the northern part (*Figure 8*), where the surface appears very damaged by cracks of different size and by plant roots. Some parts of the surface even subsided into micro-grabens developed along the main faults. Consequently, the anterior portion of the track L8/S1-6 is no longer visible because it is situated in one of these lowered parts (*Figure 3*). Moreover, a zigzag channel, probably formed by a large root, crosses the northern half of this test-pit from SE to NW, so that L8/S1-5 is virtually indiscernible (*Figure 3*). In the western portion of L8, three large rounded holes (green circles in *Figure 2*) originated from roots of acacia trees that grew on the surface. Raindrop imprints are visible to the northern edge of the test-pit (*Figure 2*) on two relatively well-preserved portions of the tuff surrounded by weathered and lowered areas. These features have also been described in several other footprint-bearing sites at Laetoli (*Leakey, 1987a*).

The situation is different in M9, where about 72 cm of grey soil and unaltered sediments overlie the Footprint Tuff. Here, the tracks are sealed by the upper, laminated part of Tuff seven and filled with strongly cemented sediment. The tuff is here in reasonably good condition, even if it is crossed

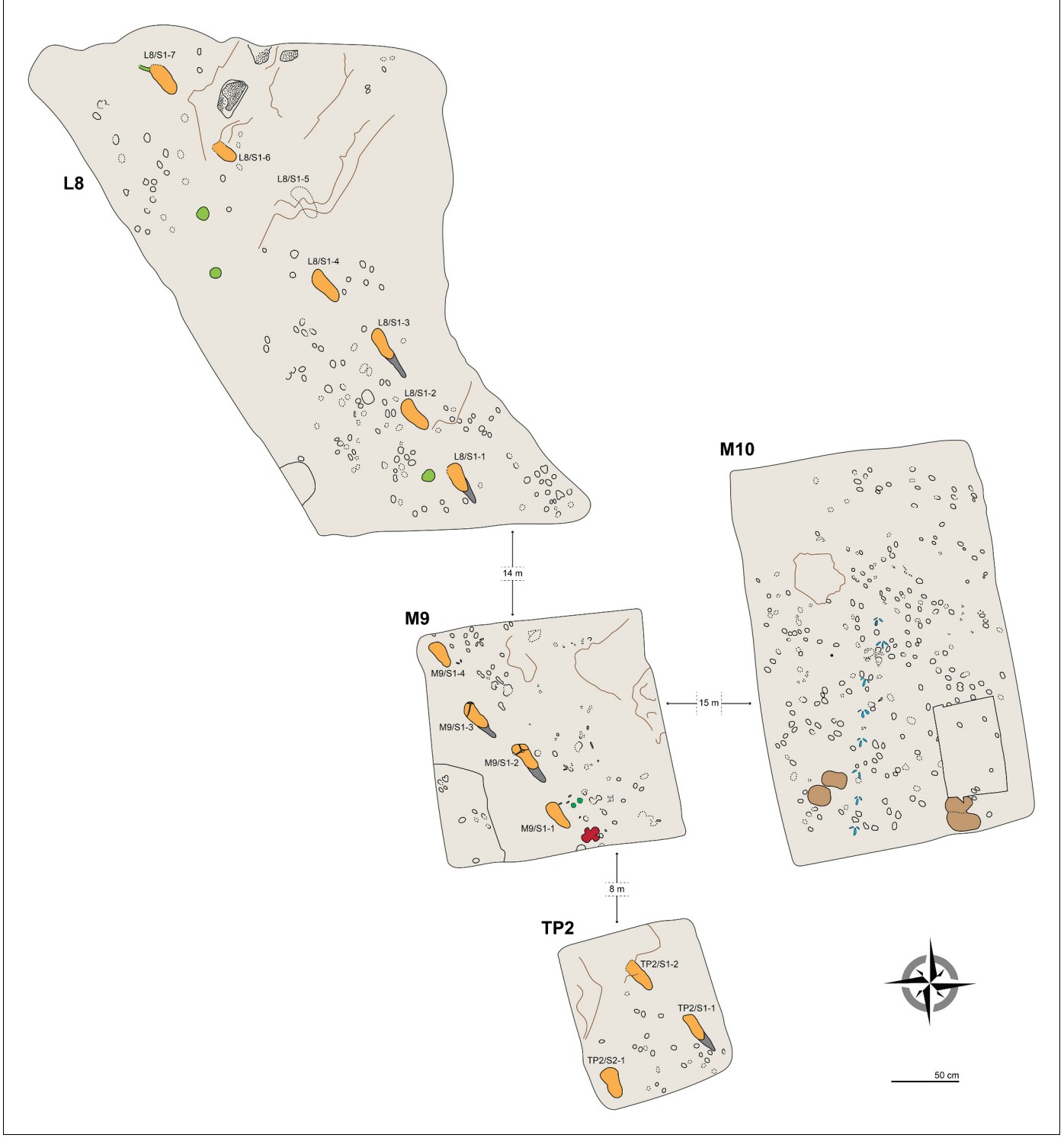

**Figure 2.** Plan view of the four test-pits excavated at Laetoli Site S. Dashed lines indicate uncertain contours. Some of the most interesting tracks are coloured: hominins in orange (heel drags in dark grey), equid in dark green (**M9**), rhinoceros in red (**M9**), giraffe in light brown (**M10**), and guineafowl in blue (**M10**). Large roots and the bases of trees are in light green (**L8**). The main faults/fractures are indicated by brown lines. Raindrop impressions occur in the northern part of L8 (dotted areas).

The following figure supplements are available for figure 2:

**Figure supplement 1.** Orthophotos of selected hominin tracks from test-pit L8 at Site S.

*Figure 2 continued on next page*

*Figure 2 continued*

**Figure supplement 2.** Orthophotos of selected hominin tracks from test-pit M9 at Site S.

**Figure supplement 3.** Orthophotos of selected tracks from test-pit M10 at Site S.

by old tectonic fractures re-cemented by calcite (*Figures 5* and *9*). Moreover, deeply expanding roots penetrate preferentially into the subhorizontal fissures situated between bedding planes, dislodging the rock and fostering carbonate dissolution.

The taphonomic state of the Footprint Tuff and of the tracks is very similar in M10, which is about 80 cm deep. In M9, the infilling matrix was removed from two hominin tracks (M9/S1-2 and M9/S1-3) (*Figures 5* and *9*) in order to examine their inner morphology. Small amounts of water were used during the excavation, in order to soften the sediment and darken its hue to better distinguish it from the surrounding tuff. The infill was finally removed by small dental tools, trying not to damage the very thin calcite film covering the original footprint surface (*White and Suwa, 1987*). Unfortunately, some vertical crisscross fractures filled by hard calcite veins (*Figures 5* and *9*) preclude a detailed morphological study of the two footprints. An about 4-cm-thick layer of tuff was removed from a footprint-free area of the M9 SW corner, putting into light a deeper horizon containing bovid tracks (*Figure 2*).

In TP2, the preservation state of the ~66-cm-deep printed tuff is intermediate between the L8 and M9/M10 ones. The southern part is in better condition: the hominin track TP2/S1-1 is rather well preserved and some of the other animal prints are still filled by the sediment of the overlying unit. Unfortunately, the SW portion of the test-pit is crossed longitudinally by north-running roots that cross TP2/S2-1, partially damaging it (*Figures 2* and *6*). On the contrary, the northern part of the test-pit is poorly preserved because of a micro-graben developed along an EW-trending fault, which also crosses TP2/S1-2, causing the lowering of its anterior portion (*Figures 2* and *6*).

## Geological setting

The assessment of the Laetoli Site S sequence within the wider framework of the Eyasi Plateau formations is crucial to understand the stratigraphic relationships between the footprint-bearing units of the newly discovered Site S and those of the historical Site G. These relationships can be discussed at two levels of increasing detail, each one affecting different and similarly more detailed aspects of the study of the tracks.

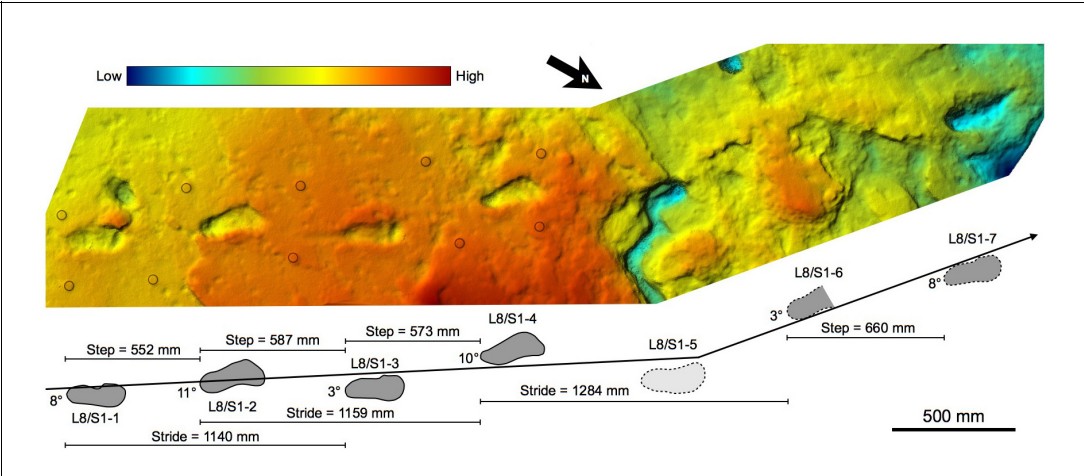

**Figure 3.** Shaded 3D photogrammetric elevation model of the L8 trackway. Colour renders heights as in the colour bar. The empty circles indicate the position of the targets of the 3D-imaging control point system (see Materials and methods for details).

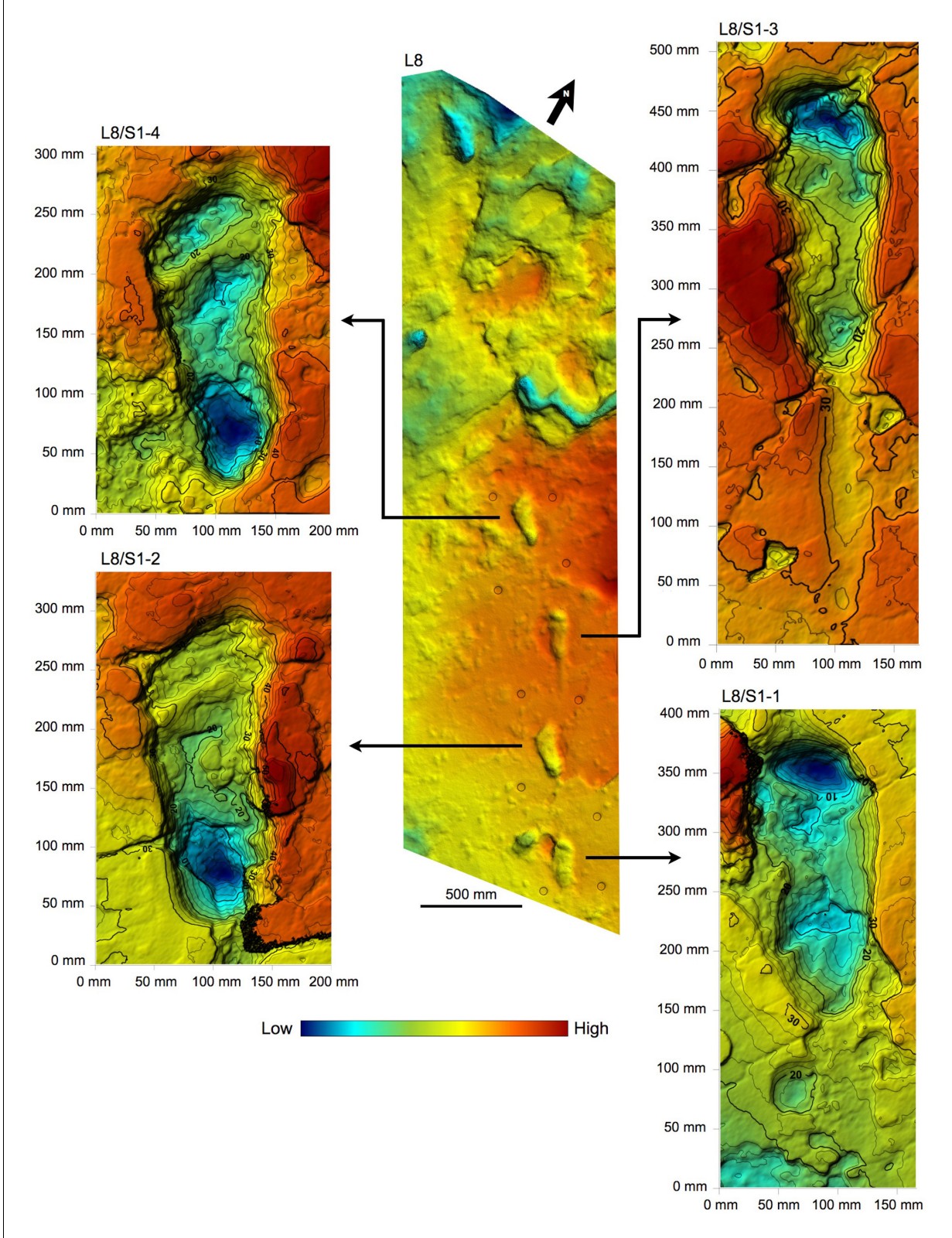

**Figure 4.** Shaded 3D photogrammetric elevation model of test-pit L8 and close-up of the best-preserved tracks with contour lines. Colour renders heights as in the colour bar; distance between elevation contour lines is 2 mm. The empty circles indicate the position of the targets.

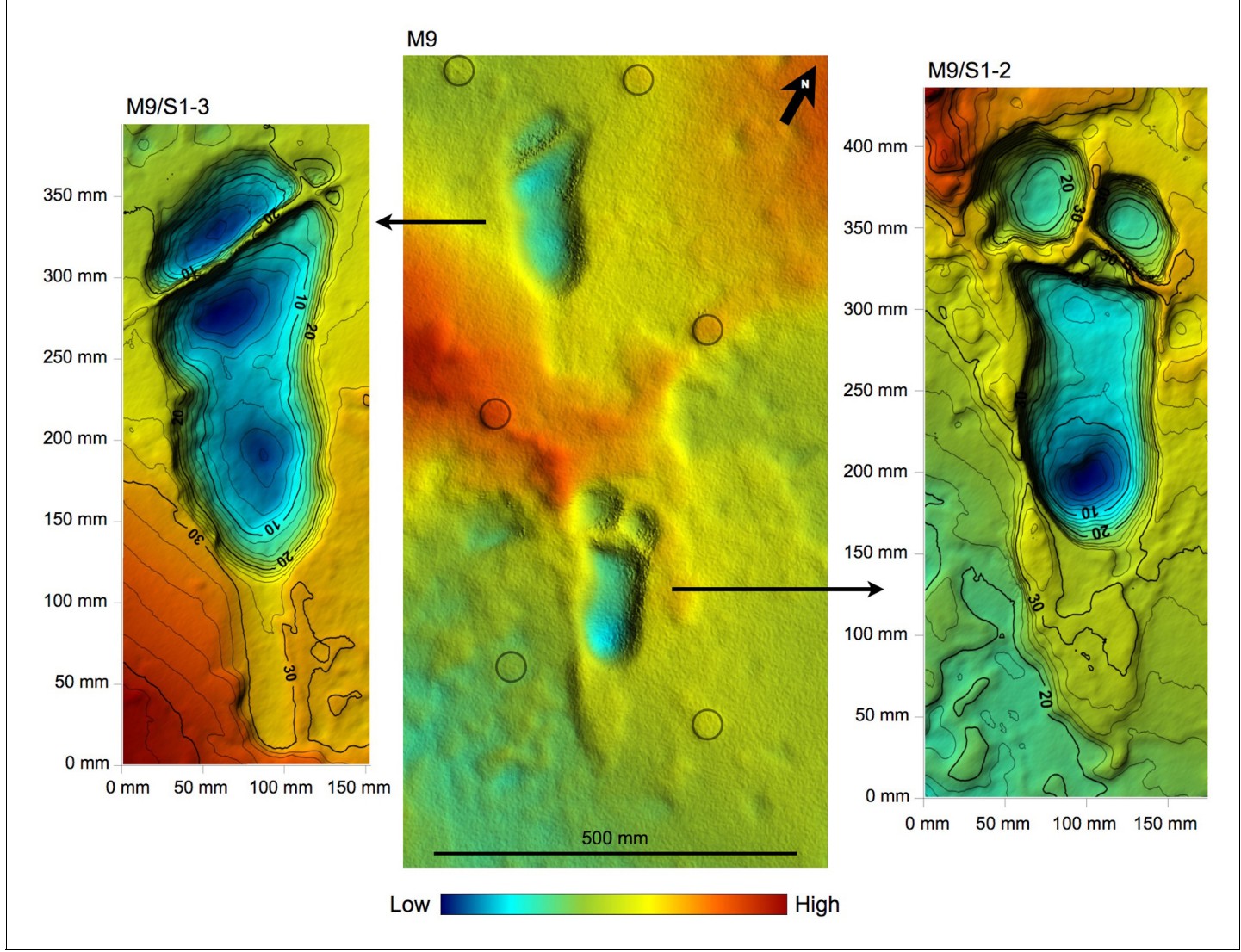

**Figure 5.** Shaded 3D photogrammetric elevation model of the central portion of test-pit M9 and close-up of the best-preserved tracks with contour lines. Colour renders heights as in the colour bar; distance between elevation contour lines is 2 mm. The empty circles indicate the position of the targets

The first – and most relevant – level regards verifying whether the unit bearing the new tracks corresponds to the Footprint Tuff, part of Tuff 7 together with the overlying Augite Biotite Tuff (*Leakey and Hay, 1979*, p. 317; *Hay, 1987*, p. 36), where the Site G tracks were printed. This would imply that the trackways are contemporaneous from a geological/geochronometric point of view. Moreover, considering that Tuff 7 includes a sequence of several sublevels originated by distinct eruptions closely spaced in time, and that its overall deposition time was estimated in weeks (*Hay and Leakey, 1982*, p. 55; *Hay, 1987*, p. 36, it can be concluded that all the tracks belong to the same general population of hominins.

Secondarily, stratigraphic relationships can be explored at higher detail, in order to assess whether the tracks of Site S were printed on exactly the same sublevel of the Footprint Tuff as those in Site G. This aspect would mostly concern the behavioural aspects of a hypothetical single group of hominins, but it must be pointed out that extra-fine correlation between outcrops, even in a depositional environment with moderate lateral variability like the Footprint Tuff deposition area, can be affected by major uncertainty.

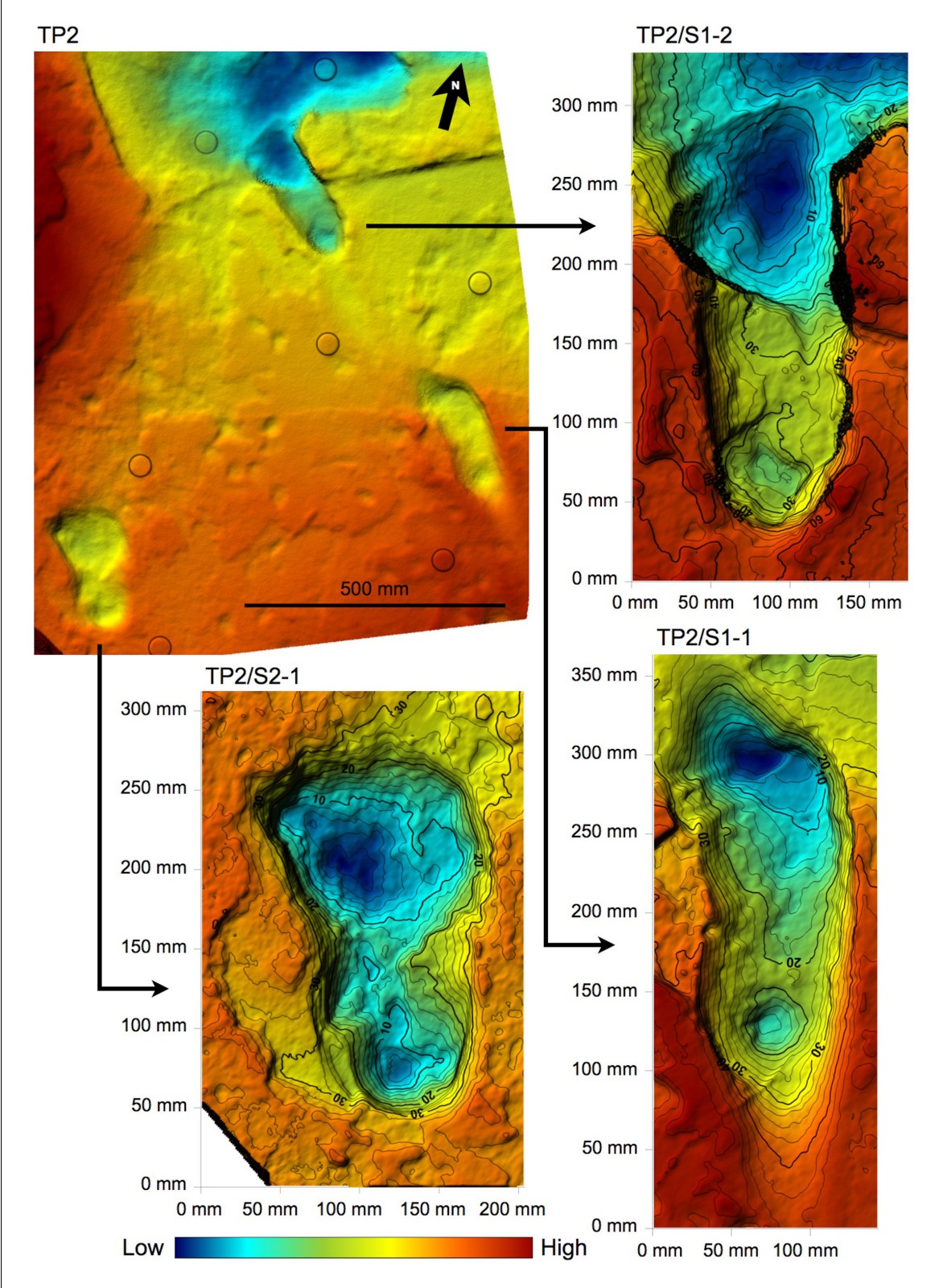

**Figure 6.** Shaded 3D photogrammetric elevation model of test-pit TP2 and close-up of the three hominin tracks with contour lines. Colour renders heights as in the colour bar; distance between elevation contour lines is 2 mm. The empty circles indicate the position of the targets.

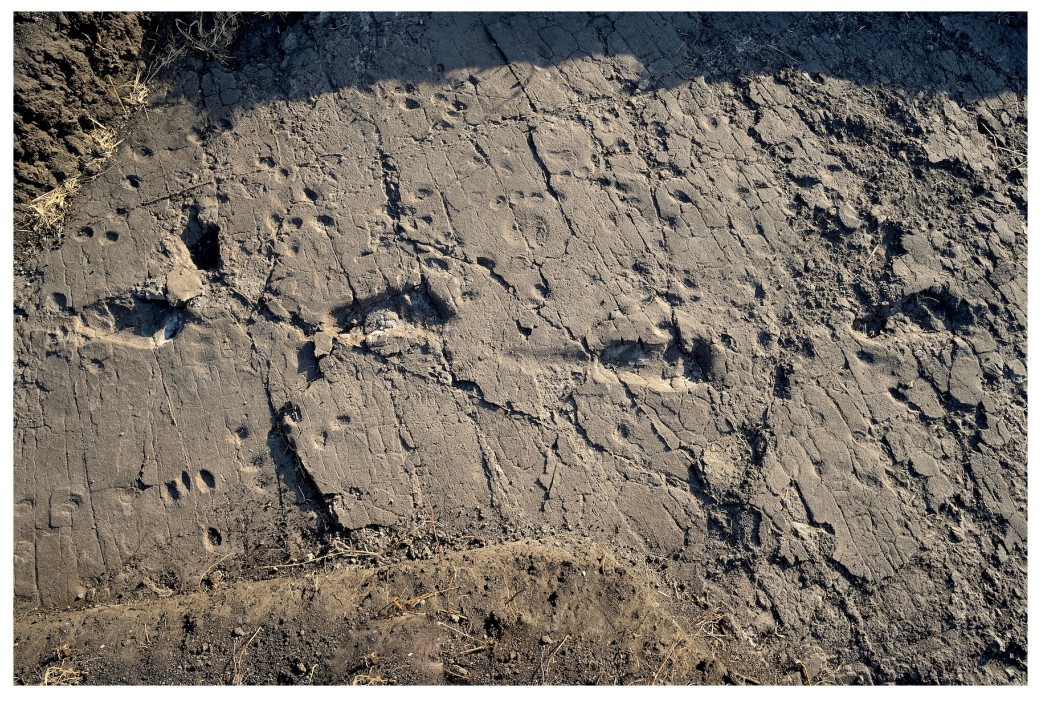

**Figure 7.** Southern part of the hominin trackway in test-pit L8. Footprints L8/S1-1, L8/S1-2, L8/S1-3 and L8/S1-4 are visible from left to right. The heel drag mark is well visible posteriorly to L8/S1-3.

## Field description of the sequences

The eye-scale characteristics of the profiles exposed in the test-pits are reported here from the top downwards.

### Test-pit L8

The Footprint Tuff is extremely shallow and partly eroded in this area, which is limited by the erosional surface of a gully side. Only the lower subunit is preserved, whereas the upper one is completely pedogenised. Consequently, the tracks are not filled-up with compact sediment but only with modern soil: dark grey (2,5Y 4/1–4/2 *dark grey-dark greyish brown*) clay loam to sandy clay loam, with well-developed coarse subangular blocky structure, extremely loose and weak. To the north, the Tuff is no longer covered by soil and crops out directly from the ground surface; the rock, already fractured by tectonic stress, is partly dislodged into decimetre-size blocklets. To the south, the Tuff is overlain by 20–25 cm of soil.

### Test-pit M9 (*Figure 10*)

1. Modern soil. Dark grey (2,5Y 4/1–4/2 *dark grey-dark greyish brown*) clay loam to sandy clay loam, with well-developed coarse subangular blocky structure, rather loose and moderately weak; sand is more common at the base, where the structure is somewhat less developed. Few coarse unsorted skeleton. Few Fe/Mn-oxide mottles. Thickness 20–25 cm; abrupt and slightly undulating limit.
2. Grey augite-rich tuff. Greyish (2.5Y 4/1–5/1 *dark grey-grey*) silty sand, poorly sorted, with common very coarse sand-size black rounded grains. Massive structure, moderately strong; no sedimentary structures. Thickness 32–35 cm; sharp subhorizontal limit, frequently marked by recent roots occupying a 0–1-cm-thick planar void. Poorly sorted very fine sand to coarse sand-size particles, including common anhedral to subhedral augite, grey rounded particles, greyish-brownish aggregates, other unidentified lithics. Light grey micro- to cryptocrystalline cement.

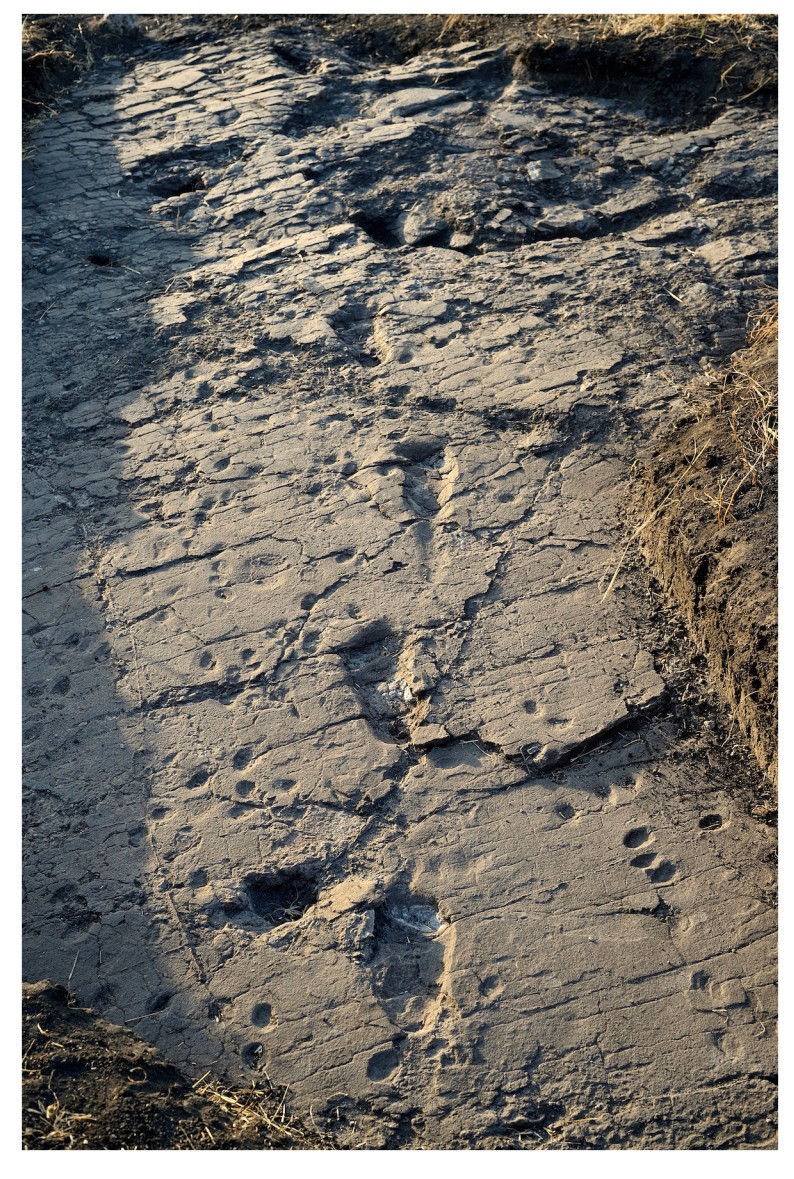

**Figure 8.** Test-pit L8 at Laetoli Site S. In the northern part of the test-pit (at the top), the Footprint Tuff is particularly altered, damaged by plant roots and dislodged along natural fractures.

3. Laminated grey tuff. Sequence of light grey to brownish to black (2.5Y 6/2 *light brownish gray*-2.5Y 5/4 *light olive brown*-N 2/5 *black*) sandy laminae and thin layers 1–3 mm thick. Massive, very strong. Thickness 5–7 cm; sharp limit marked by a fine white crust, and in some cases by a 2–5-mm-thick planar void. Moderately well-sorted anhedral to subhedral, subrounded to sub-angular, medium to fine sand-size light grey to greenish grains; white microcrystalline cement. In the uppermost layers, the grain-size is slightly coarser (medium sand), and the particles are subrounded to rounded; biotite laminae and brownish rounded aggregates are common. The darker laminae usually include finer grains, and the cement is generally less abundant.

4. Finely layered grey and white tuff. Sequence of light grey to white (N6/ *gray*-10YR 8/1 *white*) sandy layers, 2–3 mm to 25–30 mm-thick. The uppermost level is white and thicker, even if its thickness can vary significantly throughout the surface. Platy and rounded fragments of grey sediment, probably clods deriving from disarticulation of desiccation polygons, lie horizontally within the overlying white sediment. Massive, strong. Thickness 7–8 cm; sharp subhorizontal

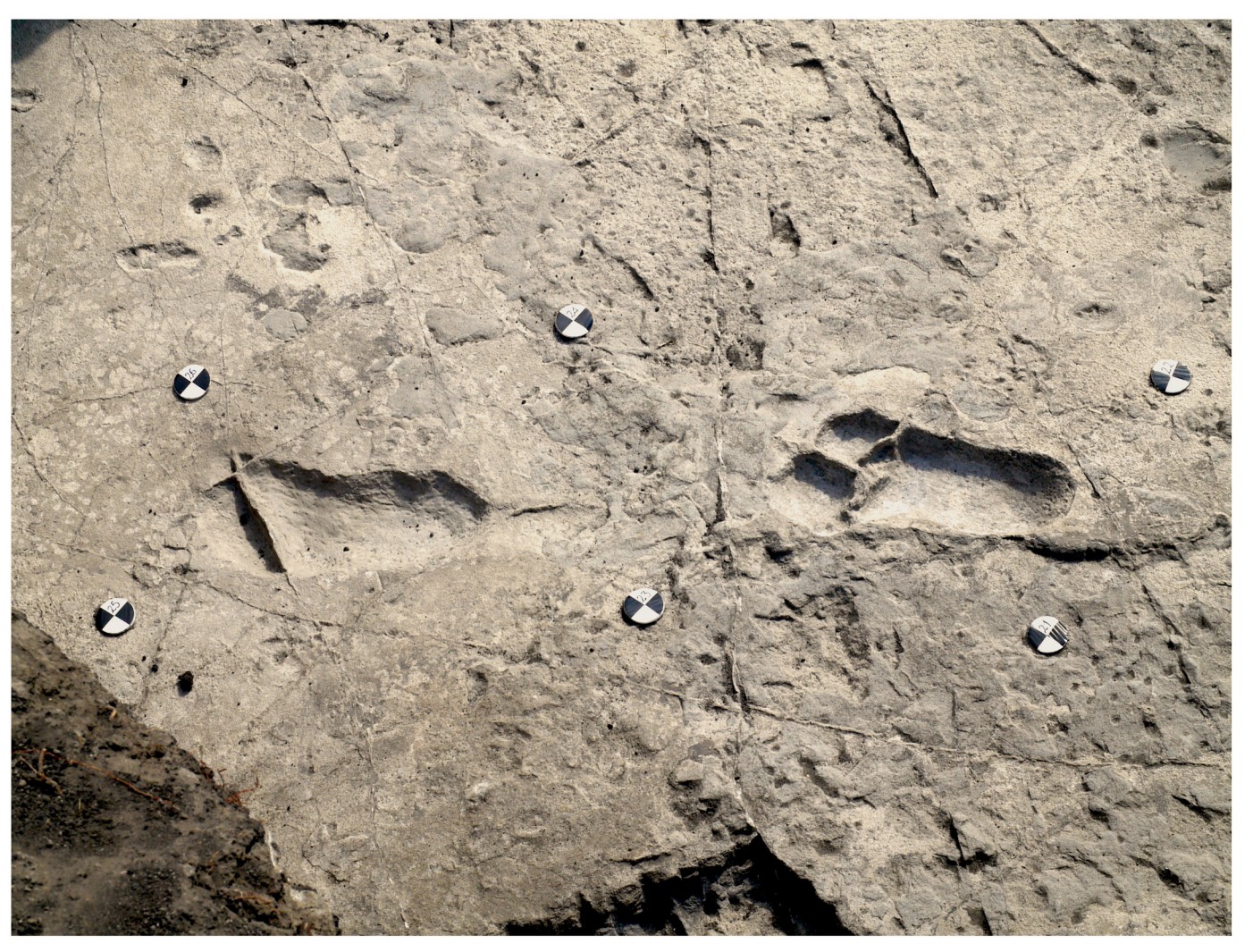

**Figure 9.** Central part of the hominin trackway in test-pit M9. Tracks M9/S1-3 and M9/S1-2 are visible from left to right. The two tracks are crossed by some fractures filled by hard calcite veins, which were not removed. In M9, the Footprint Tuff is in almost pristine condition, and most of the tracks are still filled by compact sediment.

and plain limit. Footprints at the top. The grey layers include dark grey fine sand-size particles, moderately well-sorted, rounded to subrounded, often concentrated in mm-thick laminae at the base of the layer. Some grading is not uncommon. The cement is light grey, apparently micro- or cryptocrystalline. The grains of the white layer are somewhat larger and less sorted, subrounded to angular; medium sand-size biotite laminae are frequent, as well as very light green subhedral to anhedral crystals; brownish rounded grains occur sparsely. The cement is white, apparently micro- to cryptocrystalline.

5. Light brown tuff. Homogeneous silty sand (7.5 year 6/3 *light yellowish brown*) with whitish mottles (10 year 7/1 *light gray*-5Y 8/1 *white*), poorly sorted and with common coarse sand-size rounded grains. Massive structure, very firm to moderately strong. Homogeneous, with traces of burrowers at the top. Base not observed. Very poorly sorted, silt to coarse sand-size particles, rounded to angular. Dominant grey rounded particles, frequent subhedral augite, few to frequent medium sand-size biotite laminae; rounded fragments of fine grey ash fall tuff and other still unidentified lithics occur sparsely. Whitish micro- to cryptocrystalline cement.

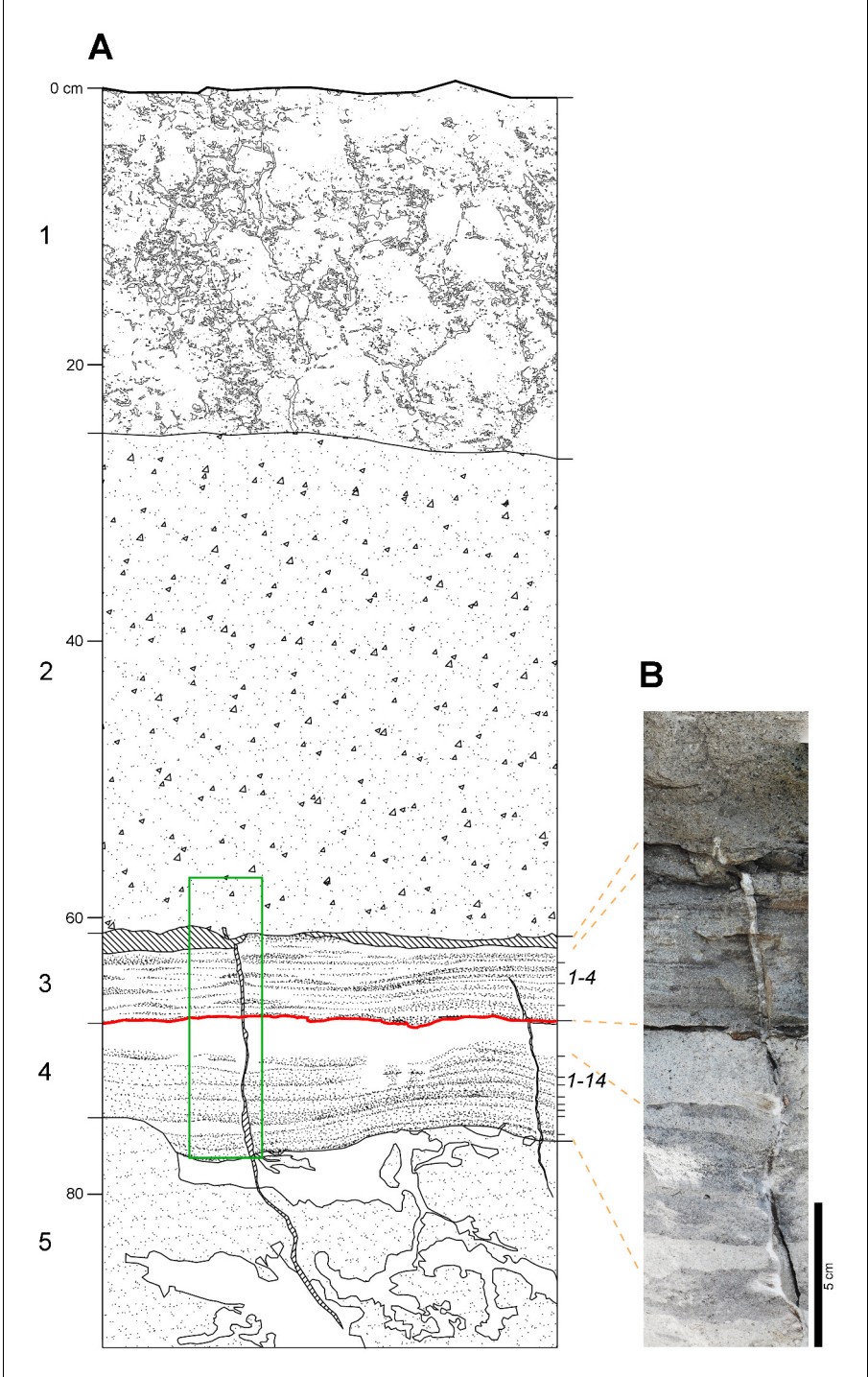

**Figure 10.** Laetoli Site S geology. (**A**) Stratigraphic sketch of the sequence, as in test-pit M9. Numbers on the left (1–5) correspond to the lithologic units observed in the field: 1 — modern soil; 2 — grey augite-rich tuff; 3 — laminated grey tuff; 4 — finely layered grey and white tuff; 5 — light brown tuff. Unit two corresponds to the Augite Biotite Tuff (*Hay, 1987*); units 3 and 4 correspond respectively to the upper and lower horizons of the Footprint Tuff (*Hay, 1987*). Numbers on the right indicate the four and fourteen sublevels included, respectively, in the upper and lower part (*Hay, 1987*). Hominin tracks occur on the topmost sublevel of unit 4 (red line); a similar thick whitish footprint-bearing level can be observed in the same stratigraphic position at Localities 6 and 7. Oblique hatch: open cracks. White patches in unit 5 are burrower tunnels and disturbances. Green rectangle: location of panel B image. (**B**) Photomosaic showing the Footprint Tuff and part of the overlying unit.

### Test-pit M10

1. Modern soil. Dark grey (2,5Y 4/1–4/2 *dark grey-dark greyish brown*) clay loam to sandy clay loam, with well-developed medium to very coarse subangular blocky structure, rather loose and moderately weak; sand is more common at the base, where the structure is somewhat less developed. Few Fe/Mn-oxide mottles. Thickness 20–45 cm; abrupt undulating limit.
2. Grey augite-rich tuff. Greyish (2.5Y 4/1–5/1 *dark grey-grey*) silty sand, poorly sorted, with common coarse to very coarse sand-size black rounded grains. Massive structure, strong; no sedimentary structures. Thickness 25–45 cm; sharp subhorizontal limit. Poorly sorted very fine sand to coarse sand-size particles, including common anhedral to subhedral augite, grey rounded particles, greyish-brownish aggregates, other unidentified lithics.
3. Laminated grey tuff. Finely interbedded light grey to brownish to black (2.5Y 6/2 *light brownish grey*-2.5Y 5/4 *light olive brown*-N 2/5 *black*) sandy laminae and thin layers 1–3 mm thick. Massive, very strong. Thickness 4–6 cm; sharp limit marked by a thin planar void. Moderately well-sorted anhedral to subhedral, subrounded to subangular, medium to fine sand-size light grey to greenish grains; white microcrystalline cement. In the uppermost layers, the grain-size is slightly coarser (medium sand), and the particles are subrounded to rounded; biotite laminae and brownish rounded aggregates are common. The darker laminae usually include finer grains, and the cement is generally less abundant.
4. Finely layered grey and white tuff. Only the top surface was observed. Common animal tracks.

### Test-pit TP2

1. Modern soil. Dark grey (2,5Y 4/1–4/2 *dark grey-dark greyish brown*) clay loam to sandy clay loam, with well-developed fine to very coarse subangular blocky structure, loose and moderately weak. Few Fe/Mn-oxide mottles. Thickness 35–45 cm; abrupt undulating limit.
2. Grey augite-rich tuff. Greyish (2.5Y 4/1–5/1 *dark grey-grey*) silty sand, poorly sorted, with common coarse to very coarse sand-size black rounded grains. Massive structure, strong; no sedimentary structures. Thickness 6–23 cm; sharp subhorizontal limit. Poorly sorted very fine sand to coarse sand-size particles, including common anhedral to subhedral augite, grey rounded particles, greyish-brownish aggregates, other unidentified lithics.
3. Laminated grey tuff. Finely interbedded light grey to brownish to black (2.5Y 6/2 *light brownish grey*-2.5Y 5/4 *light olive brown*-N 2/5 *black*) sandy laminae and thin layers 1–3 mm thick. Massive, very strong. Thickness 4–5 cm; sharp limit marked by a thin planar void. Moderately well-sorted anhedral to subhedal, subrounded to subangular, medium to fine sand-size light grey to greenish grains; white microcrystalline cement. In the uppermost layers, the grain-size is slightly coarser (medium sand), and the particles are subrounded to rounded; biotite laminae and brownish rounded aggregates are common. The darker laminae usually include finer grains, and the cement is generally less abundant.
4. Finely layered grey and white tuff. Only the top surface was observed. Common animal and three hominin tracks.

## Results

### Non-hominin tracks

Tracks and trackways of mammals, birds and insects, as well as raindrop impressions, are recorded from 18 sites at Laetoli, named alphabetically from A to R. Sites from A to P were listed and geographically located by *Leakey (1987b)*, who also described in detail the ichnological record of the most important exposures. Sites Q and R were discovered and described by *Musiba et al. (2008)*. More than 11,300 single footprints are recorded from Sites A–R. These tracks testify to a very rich ichnofauna, although a very high percentage of them (more than 88%) can be ascribed to small mammals such as lagomorphs and/or *Madoqua*-like bovids (*Leakey, 1987a*; *Musiba et al., 2008*).

Numerous footprints were discovered in the new exposures (test-pits L8, M9, TP2 and M10) of the Footprint Tuff at Site S in Locality 8 (*Figure 2*). A total of 529 footprints of mammals (excluding hominins) and birds (*Table 1*) were recorded during the September 2015 field season. The prints were carefully cleaned using soft brushes to reveal detailed features, measured, photographed, traced, mapped and identified in a preliminary study.

Mammal tracks – mostly of small and medium-size bovids – are very abundant in M10, L8 and M9 and occur less frequently in TP2. Their size (30–40 mm long and 20–36 mm wide) and morphological

**Table 1.** Number of individual tracks (excluding hominins) at Laetoli Site S.

| Taxon | L8 | M9 | TP2 | M10 | Total |
|---|---|---|---|---|---|
| Numididae (?*Numida*) | - | 4 | - | 9 | 13 |
| Bovidae, small size (?*Madoqua*) | 107 | 39 | 16 | 211 | 373 |
| Bovidae, medium size (?*Gazella*) | 39 | 9 | - | 21 | 79 |
| Equidae (?*Hipparion*) | 1 | 2 | - | - | 3 |
| Giraffidae | - | - | - | 4 | 4 |
| Lagomorpha (?*Lepus*) | 8 | - | - | 4 | 12 |
| Rhinocerotidae | - | 1 | - | - | 1 |
| Unidentified micromammals | - | 27 | - | 17 | 44 |
| Total | 155 | 82 | 26 | 266 | 529 |

features suggest that most of them can be ascribed to the genus *Madoqua* (*Figure 2* and *Figure 2—figure supplement 3*). Some slightly larger prints (60–80 × 40–60 mm) can be referred to medium-sized bovids such as *Gazella*, *Eudorcas* or *Nanger*.

It is very difficult to distinguish the footprints of *Madoqua*-like bovids from lagomorph footprints because of their very similar morphology and size (*Leakey, 1987a*). Consequently, we decided to ascribe to Lagomorpha only trails that clearly include at least four footprints arranged in the normal hare gait pattern, i.e. two single prints left by the front feet followed by a couple of prints made by the hind feet in the direction of gait. Each single trail (i.e., four footprints) is approximately 200 mm long and 100 mm wide.

We identified very few prints of giraffids (about 170 × 125 mm) in M10, equids (about 50–95 × 45–70 mm) in L8 and M9 and rhinoceroses (about 150–135 mm) in M9 (*Figure 2* and *Figure 2—figure supplement 3C*). In M9 and M10, some avian prints (about 60 × 75 mm) often organised in trails, can be referred to Galliformes of the family Numididae, such as the guinea fowl (genus *Numida*) (*Figure 2* and *Figure 2—figure supplement 3A,B*). Finally, we report some very small (about 10 × 10 mm) tracks of unidentified animals, probably micromammals, in M9 and M10.

The above-mentioned assemblage of terrestrial mammal and bird footprints suggests that the local palaeoenvironment was characterised by a mosaic of dry tropical bushland, woodland, open grassland and riverine forest similar to the extant one.

## Morphology of hominin tracks

The morphology of the S1 tracks can be described in detail, but unfortunately the only preserved track of S2 shows an abnormal widening of the anterior part. This enlarged morphology is possibly due to a lateral slipping of the foot before the toe-off; alternatively, it could be due to taphonomic factors as a thick root crossing the footprint longitudinally may have altered its original morphology. The overall morphology of the S1 tracks matches those at Site G (*Figure 11*) and is similar in particular to the prints of the larger individual, G2 (*Robbins, 1987*): the heel has an oval shape and is pressed deeply into the ground; the medial side of the arch is higher than the lateral one; the ball region is oriented at an angle of about 75° with respect to the longitudinal axis of the foot and is delimited anteriorly by a transversal ridge, formed when the toes gripped the wet ash and pushed it posteriorly. No clear distinction among the toes is visible. The adducted hallux extends more anteriorly than the other toes in all visible footprints. In TP2/S1-1, the hallux apparently shuffled anteriorly when the foot was lifted from the ground. Some tracks (especially L8/S1-3, M9/S1-2, M9/S1-3 and TP2/S1-1) are characterised by a posterior drag mark about 100 mm long (*Figures 4–7* and *Figure 2—figure supplements 1* and *2*). These marks were possibly left by the heel shuffling on the ash before being firmly placed into the soil. The two latter features were also recognised in some of the G2 prints (*Robbins, 1987*) and suggest that the feet were probably lifted above the ground at a low oblique angle. The depth distribution pattern indicates that the weight transfer of S1 was similar to that described for G1–3 (*Robbins, 1987*): starting from the heel, the weight was transferred along the lateral part of the foot (note the steep slope of the lateral wall of the tracks compared to that on

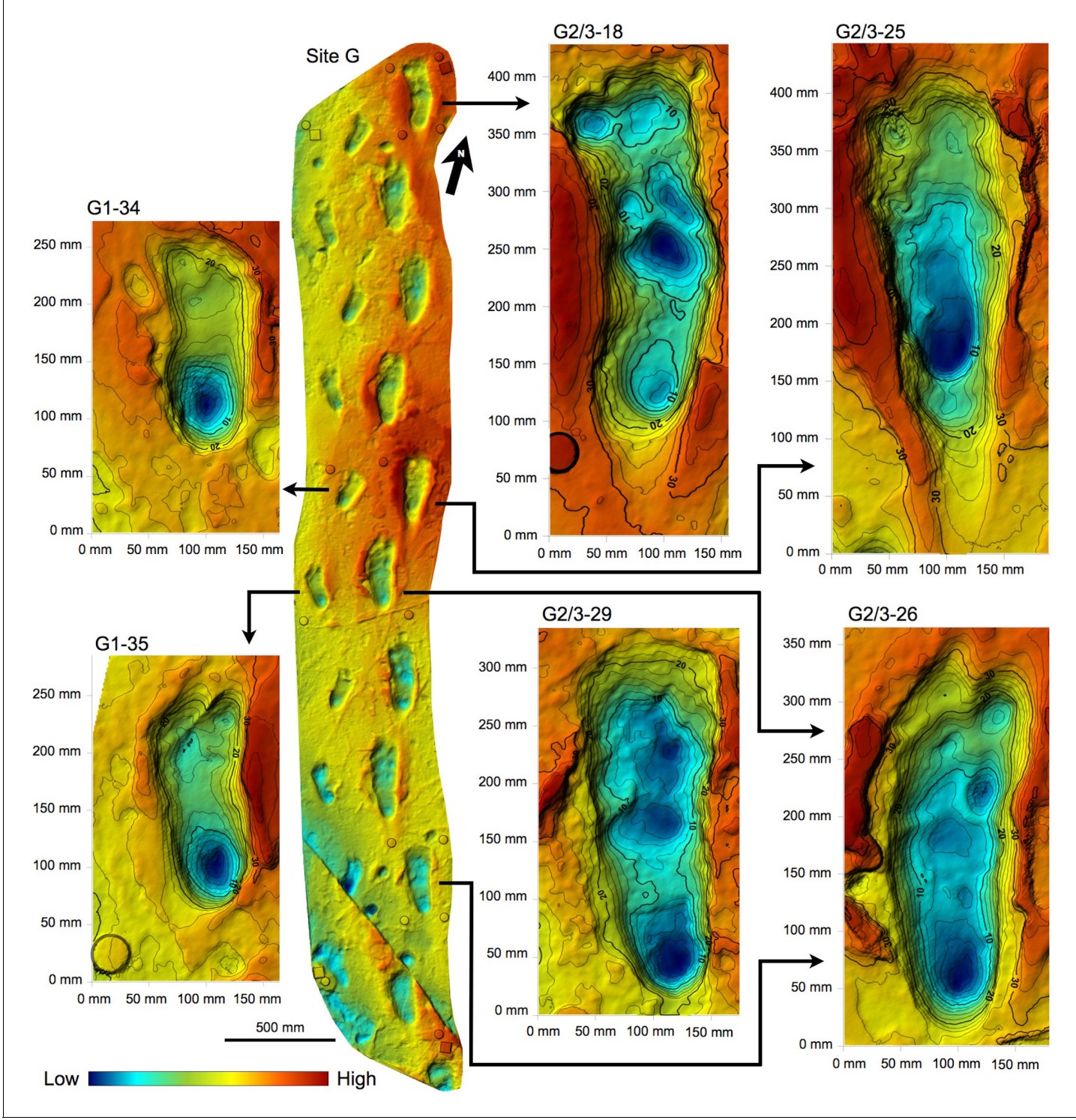

**Figure 11.** Shaded 3D photogrammetric elevation model of a cast of the southern portion of the Site G trackway with close-ups of selected hominin tracks with contour lines. Colour renders heights as in the colour bar; distance between elevation contour lines is 2 mm. The empty circles and squares indicate the position of the targets.

The following figure supplement is available for figure 11:

**Figure supplement 1.** Orthophotos of selected hominin footprints from a cast of the southern portion of the Site G trackway.

the medial side) up to the distal metatarsal region, and from here to the toes. In some of the S1 tracks (L8/S1-1, L8/S1-3 and TP2/S1-8, all of the right side), however, the area of maximum depth is located beneath toes 2–5. This may suggest a somewhat asymmetrical walking, in which the weight was sometimes loaded on the anterolateral part of the foot before the toe-off. Alternatively, this pattern may be indicative of a rotation of the upper body during the gait (*Schmid, 2004*). The angle of gait ranges approximately from 2° to 11°, without any particular difference between the right and left sides. Regarding this aspect, S1 resembles more G2/3, for which very low average angles are reported, whereas G1 shows instead wide asymmetrical angles (*Tuttle, 1987*).

## Speed, stature and body mass estimates

The main dimensional parameters of the tracks at Site S are presented in *Table 2* (the single measurements are explained in Materials and methods).

Speed estimates for S1 and G1–3 were computed starting from stride length (*Figure 3*) (see Materials and methods). The obtained values (*Table 3*) show that these hominins were all walking at similar low speed (about 0.44 to 0.9 m/s, depending on the analysis method).

The average length of the tracks in the S1 trackway is 261 mm (range 245–274). Lower values were measured for the three individuals at Site G. The average lengths are 180 mm for G1, 225 mm for G2 and 209 mm for G3 (*Leakey, 1981*; *Tuttle, 1987*) (*Table 3*), although a digital analysis-based study (*Bennett et al., 2016*) of some Site G footprint casts suggests higher values for G1 (193 mm) and G3 (228 mm). The main metrical features of the S1 and S2 tracks (footprint length and width, step and stride lengths) are larger than the G1–3 equivalents (*Table 3*).

The stature and mass of the Laetoli print-makers were estimated following the relationships between foot/footprint size and body dimensions (*Tuttle, 1987*; *Dingwall et al., 2013*). It must be pointed out that stature and body-mass estimates obtained by linear regressions from modern humans (*Tuttle, 1987*; first method by *Dingwall et al., 2013* are probably exaggerations, as the body proportions of modern *Homo sapiens* are considerably different from those of the Laetoli putative track-makers. Consequently, we focused our interpretations on the more appropriate predictions inferred from the relationship between foot size and body dimensions in *Australopithecus* (second method by (*Dingwall et al., 2013*; see Materials and methods for details). The data in *Tables 2–3* indicate that stature and mass estimates for S1 and S2 (about 165 cm and 44.7 kg, and 146 cm and 39.5 kg, respectively) are higher than those obtained for G1, G2 and G3 (with S2 partly overlapping the higher estimates for G2).

## Discussion

### Stratigraphic position of the new tracks

Site S is situated on an almost level or very gently dipping surface, situated at the foot of the left (southern) side of the Garusi River valley. Site G is situated about 150 m to the north, on the same surface but 1.5–2 m lower than Site S. Several shallow gullies dissect this surface, producing a complexly terraced morphology: consequently, there is no observable stratigraphic continuity between the two sites. However, the gullies put into light about 2–3 m of the underlying sequence, whose units are horizontally layered and characterised by almost constant thickness. Only a shallow depression elongated E-W can be observed between the sites; this is probably an ancient erosion channel filled by a constant thickness of the Site S footprint-bearing tuff. Even if the area of possible outcrop of the Footprint Tuff on gully sides close to Site S is covered by debris, the correlation between G and S is in general straightforward.

All previous literature describing the original stratigraphic setting at Laetoli (*Leakey and Hay, 1979*; *Hay and Leakey, 1982*; *Hay, 1987*) indicates that the Footprint Tuff can be divided into two main units – the lower and the upper one – which can be subdivided into 14 and 4 sublevels, respectively. Footprints occur on several sublevels of each unit all over the Laetoli area: eight within the lower one (mostly on sublevel 9 and on the topmost sublevel 14), and two within the upper one (sublevels 1 and 2).

*Leakey and Hay (1979*, pp. 317–318 and fig. 4) provided a brief description of the type-sequence of the Footprint Tuff at Locality 6 (Site A), where a short trackway of human-like footprints – later referred to an ursid (*Tuttle, 2008*) – was also found. Later, *Hay and Leakey (1982*, p. 55 and

**Table 2.** Dimensional parameters measured and derived from the Laetoli Site S tracks and stature and body mass estimates for S1 and S2.

| Footprint | Side | Length (mm) | Max width (mm) | Foot index (%) | Heel width (mm) | Angle of gait (degrees) | Estimated stature (cm) | | | Estimated body mass (kg) | |
|---|---|---|---|---|---|---|---|---|---|---|---|
| | | | | | | | H. sapiens[§] | H. sapiens[°] | Au. afarensis[‡] | H. sapiens[°] | Au. afarensis[‡] |
| TP2/S1-1 | right | 271 | 101 | 37.2 | 83 | 6 | 194–170 | 175.4 | 167–175 | 53.8 | 42.9–50.0 |
| TP2/S1-2 | left | 271 | 99 | 36.6 | 81 | 4 | 193–169 | 175.1 | 167–175 | 53.1 | 42.8–49.8 |
| M9/S1-1 | left | 250 | 102 | 40.6 | 73 | 2 | 179–156 | 167.5 | 154–161 | 51.6 | 39.6–46.0 |
| M9/S1-2 | right | 264 | 105 | 39.7 | 80 | 3 | 189–165 | 172.8 | 163–171 | 54.2 | 41.8–48.7 |
| M9/S1-3 | left | 268 | 111 | 41.2 | 91 | 4 | 192–168 | 174.3 | 166–173 | 56.3 | 42.5–49.4 |
| M9/S1-4 | right | 245 | 101 | 41.2 | 71 | 4 | 175–153 | 165.6 | 151–158 | 50.9 | 38.8–45.1 |
| L8/S1-1 | right | 245 | 104 | 42.4 | 78 | 8 | 175–153 | 165.6 | 151–158 | 51.7 | 38.8–45.1 |
| L8/S1-2 | left | 265 | 106 | 40.0 | 82 | 11 | 189–166 | 173.1 | 164–171 | 54.5 | 41.9–48.8 |
| L8/S1-3 | right | 260 | 103 | 39.6 | 77 | 3 | 186–163 | 171.3 | 161–168 | 53.1 | 41.2–47.9 |
| L8/S1-4 | left | 274 | 106 | 38.6 | 81 | 10 | 196–171 | 176.5 | 169–177 | 55.6 | 43.4–50.5 |
| L8/S1-5 | right | - | - | - | - | - | - | - | - | - | - |
| L8/S1-6 | left | - | - | - | 86 | 3 | - | - | - | - | - |
| L8/S1-7 | right | 258 | 110 | 42.7 | 90 | 8 | 184–161 | 170.3 | 159–166 | 54.8 | 40.7–47.4 |
| **Average S1** | - | **261** | **104** | **40.0** | **81** | **6** | **184–163** | **171.6** | **161–168** | **53.6** | **41.3–48.1** |
| TP2/S2-1 | right | 231 | 120* | 51.9* | 86 | - | 165–144 | 160 | 142–149 | 46.7 | 36.5–42.4 |

**Step length**

| Footprints | Side | Step length (mm) |
|---|---|---|
| TP2/S1-1 → 2 | right → left | 553 |
| M9/S1-1 → 2 | left → right | 548 |
| M9/S1-2 → 3 | right → left | 505 |
| M9/S1-3 → 4 | left → right | 571 |
| L8/S1-1 → 2 | right → left | 552 |
| L8/S1-2 → 3 | left → right | 587 |
| L8/S1-3 → 4 | right → left | 573 |
| L8/S1-6 → 7 | left → right | 660 |
| Average right → left | | 545 |
| Average left → right | | 591 |
| Average | | 568 |

**Stride length**

| Footprints | Side | Stride length (mm) |
|---|---|---|
| M9/S1-1 → 3 | left | 1044 |
| M9/S1-2 → 4 | right | 1069 |
| L8/S1-1 → 3 | right | 1140 |
| L8/S1-2 → 4 | left | 1159 |
| L8/S1-4 → 6 | left | 1284 |
| Average right | | 1105 |
| Average left | | 1162 |
| Average | | 1139 |

*Values overestimated because of the enlarged morphology of the only preserved track of S2. [§]Estimation based on footprint length and stature in *Homo sapiens* (**Tuttle, 1987**). [°]Estimation based on the relationship between footprint length and stature/body mass in *H. sapiens* (**Dingwall et al., 2013**). [‡]Estimation based on the relationship between foot length and stature/body mass in *Au. afarensis* (**Dingwall et al., 2013**). See Materials and methods for details.

*White and Suwa (1987*, p. 488 specified that the hominin tracks at Site G are situated on the top of horizon B, i.e. on the top of sublevel 14 within the lower unit of the Footprint Tuff. Eventually, *Hay (1987*, pp. 34–35 and fig. 6) provided a generalised columnar profile of the Footprint Tuff; this is by far the most accurate description available, but is averaged over all the Laetoli area sites. Although the stratigraphic descriptions above are very accurate, they do not provide details about the eye-scale characteristics of the tuffs, i.e. colour, texture, limits, and so on, and no photographs of the sequence have been published.

The Site S sequence does not fit the aforementioned descriptions perfectly, at least not within the observed area, which is rather narrow. The grey augite-rich tuff of Site S largely matches the description of the Augite Biotite Tuff described by *Hay (1987*, p. 34 and following, level 4 in fig. 2.6, p. 35). Regarding the Footprint Tuff, the upper unit corresponds to Site S Laminated Grey Tuff, but the sublevels here are layered rather crudely, whereas the most evident sedimentary structure is a very fine and almost continuous lamination, which makes the subdivision rather problematic. Energy-sorting of denser grains is apparently a relevant aspect of the depositional processes. The Finely Layered Grey and White Tuff of Site S corresponds to the lower subunit of the Footprint Tuff; 14 sublevels are apparent as in the standard description, but this number may be imprecise (or evaluated differently) because some of them are extremely thin and apparently discontinuous; in fact, some of the thinner (and darker) ones look more like concentrations of gravity-sorted coarser/denser grains situated at the bottom of graded layers. The top sublevel is rather thicker than the others and somewhat whitish instead of greyish, as apparent also in Localities 6 and 7.

Some lateral variability is not surprising in continental environments, which are normally affected by strong morphogenetic processes and/or lateral changes in the sedimentary environments. Consequently, lateral variability can also be expected within the sequence of the Footprint Tuff, even if the involved volcanic depositional processes were rather uniform over a wide area around Laetoli and gave the whole sequence a remarkably homogeneous aspect throughout its outcrops.

The correlation between Site G and Site S cannot be absolutely undisputable, at least for the time being, because the original profile could not be examined directly. However, the geological and morphological setting of the area, as well as the characteristics of the newly exposed sequence, indicate with a very good margin of confidence that the newly discovered tracks belong to the Footprint Tuff.

To provide a more accurate correlation within the Footprint Tuff, we observe that the Site S tracks were printed on the uppermost level of the Finely Layered Grey and White Tuff (unit 4 in the description provided in this paper), which corresponds to the lower subunit of the Footprint Tuff. The lithological change to the overlying subunit is very evident and marked by a sharp surface, often underlined by a thin crack. However, because of the aforementioned dissimilarities, it is not possible to assess with reasonable confidence whether this stratigraphic position also corresponds to the top of level 14 in the standard sequence (*Hay, 1987*, p. 35, fig. 2.6), i.e. to the same stratigraphic position as the Site G trackways.

## Implications of the new Laetoli footprints

Our results show that no matter which method is employed to estimate stature and body mass (see Material and methods), the two individuals S1 and S2 were taller and had a larger body mass than the G individuals. The estimated about 165 cm stature of S1 is quite remarkable, exceeding G2 by more than 20 cm (*Table 3*).

In order to contextualise the australopithecine and early *Homo* stature estimates and range of variability obtained from the footprints within a broader picture (*Figure 12*), and to compare them with a larger sample, we extended our analysis to consistent data based on skeletal elements, namely femurs (see Materials and methods for details). *Figure 12* shows the estimated stature of australopithecine and early *Homo* individuals by species between 4.0 and 1.0 Ma. The predicted stature of S1 exceeds any australopithecine: a mean value of 158 cm was estimated for the large *Au. afarensis* individual from Woranso-Mille (*Haile-Selassie et al., 2010*; *Lovejoy et al., 2016*), while the Hadar individuals range from 109 to 143 cm (*McHenry, 1991*; *Ward et al., 2012*) (*Figure 12*). The stature of S1 falls within the range of modern *Homo sapiens* maximum values; it also fits the available *Homo erectus sensu lato* estimates based on fossil remains (*Ruff and Walker, 1993*) and on footprints (*Bennett et al., 2009*) (*Figure 12*). At the same time, the 41 to 48 kg body mass range estimated for S1 (*Table 3*) falls easily within the range of male *Au. afarensis* (40.2–61.0 kg)

**Table 3.** Data and estimates for the five Laetoli track-makers from Sites S and G. Limited to S1, mean values, standard deviation and range are provided.

| Trackway | | S1 | S2 | G1 | G2 | G3 |
|---|---|---|---|---|---|---|
| Number of measurable footprints | | 11 | 1 | 9 | 2 | 8 |
| Average footprint length (mm) | | 261 ± 10.5 (245–273) | 231 | 180 | 225 | 209 |
| Average footprint max width (mm) | | 104 ± 3.7 (99–111) | 120[*] | 79 | 117 | 85 |
| Average foot index (%) | | 40.0 ± 1.9 (36.6–42.7) | 51.9[*] | 43.8 | 48.0 | 41.5 |
| Average step length (mm) | | 568 ± 44.3 (505–660) | - | 416 | 453 | 433 |
| Average stride length (mm) | | 1139 ± 94.0 (1044–1284) | - | 829 | 880 | 876 |
| Estimated stature (cm) | *H. sapiens*[§] | 163–186 | 144–165 | 113–129 | 141–161 | 130–149 |
| | *H. sapiens*[°] | 171.6 ± 5.4 | 160 ± 5.4 | 141.4 ± 5.4 | 158.2 ± 5.4 | 152.2 ± 5.4 |
| | *Au. afarensis*[‡] | 161–168 | 142–149 | 111–116 | 139–145 | 129–135 |
| Estimated body mass (kg) | *H. sapiens*[°] | 53.6 ± 3.7 | 46.7 ± 3.8 | 39.3 ± 3.7 | 52.6 ± 3.7 | 43.2 ± 3.7 |
| | *Au. afarensis*[‡] | 41.3–48.1 | 36.5–42.4 | 28.5–33.1 | 35.6–41.4 | 33.1–38.5 |
| Walking speed (m/s) | | 0.47–0.55 (0.93) | – | 0.43–0.50 (1.00) | 0.36–0.42 (0.79) | 0.39–0.46 (0.88) |
| Relative speed ($s^{-1}$) | | 0.25–0.34 (0.54) | – | 0.33–0.44 (0.71) | 0.23–0.30 (0.50) | 0.26–0.35 (0.58) |

[*]Values overestimated because of the enlarged morphology of the only preserved track of S2. [§]As in **Table 2**. [°]As in **Table 2**. [‡] As in **Table 2**. For walking speed and relative speed, values outside the brackets are based on the method of **Alexander (1976)**, those inside the brackets are based on the method of **Dingwall et al. (2013)**. See Materials and methods for details.

(*Grabowski et al., 2015*). These results extend the dimensional range of the Laetoli track-makers and identify S1 as a large-size individual, probably a male (*Plavcan, 1994*; *Grabowski et al., 2015*).

These findings provide independent evidence for large body-size individuals among hominins as ancient as 3.66 Ma. Consequently, we may emphasise the conclusions by *Grabowski et al. (2015)* and *Jungers et al. (2016)*, who reported that the body sizes of the australopithecines and of the early *Homo* representatives were similar, but also that certain australopithecine individuals (at least of *Au. afarensis*) were comparable with later *Homo* species, including *H. erectus s. l.* and *H. sapiens*. Thus, our results support a nonlinear evolutionary trend in hominin body size (*Di Vincenzo et al., 2015*; *Jungers et al., 2016*) and contrast with the idea that the emergence of the genus *Homo* and/ or the first dispersal out of Africa was related to an abrupt increase in body size (*McHenry and Coffing, 2000*; *Antón et al., 2014*; *Maslin et al., 2015*). The identification of large-size individuals among the australopithecines – i.e. hominins commonly presumed to be small-bodied on average – shows also that the available fossil record can be misleading, resulting in an underestimate of the hominin phenotypic diversity in any given period.

Moreover, ascribing the S1 tracks to a possible male requires that we reconsider the sex and age of the other Laetoli individuals, who have been object of a plethora of interpretations (and associated illustrations largely disseminated to the public) since Mary Leakey's work (*Leakey, 1981*). The most parsimonious option is that sex and age of the hominins represented at Site G cannot be determined, as subadult individuals could possibly be present among them. However, the body-mass estimates suggest some observations as G1 and G3 fall within the range of putative *Au. afarensis* females (25.5–38.1 kg, according to *Grabowski et al. [2015]*), whereas G2 and S2 span across the upper female and the lower male ranges (40.2–61.0 kg, according to *Grabowski et al. [2015]*) . All of these individuals are definitively smaller than the body mass calculated from the S1 tracks. A possible tentative conclusion is that the various individuals represented at Laetoli are: S1, a male; G2 and S2, females; G1 and G3, smaller females or juvenile individuals.

Evidence for either marked or moderate body-size variation in *Au. afarensis*, based on data collected in a single site, was limited until now to the fossil assemblage from the Hadar 333 locality, dated to 3.2 Ma (with body masses ranging from 24.5 to 63.6 kg). The new estimates for the Laetoli individuals indicate an even more marked variation in body size within the same hominin population, at 3.66 Ma. Consequently, the combined records from Laetoli and Hadar suggest that large-bodied hominins existed in the African Pliocene for over 400,000 years, between 3.66 and 3.2 Ma. At the

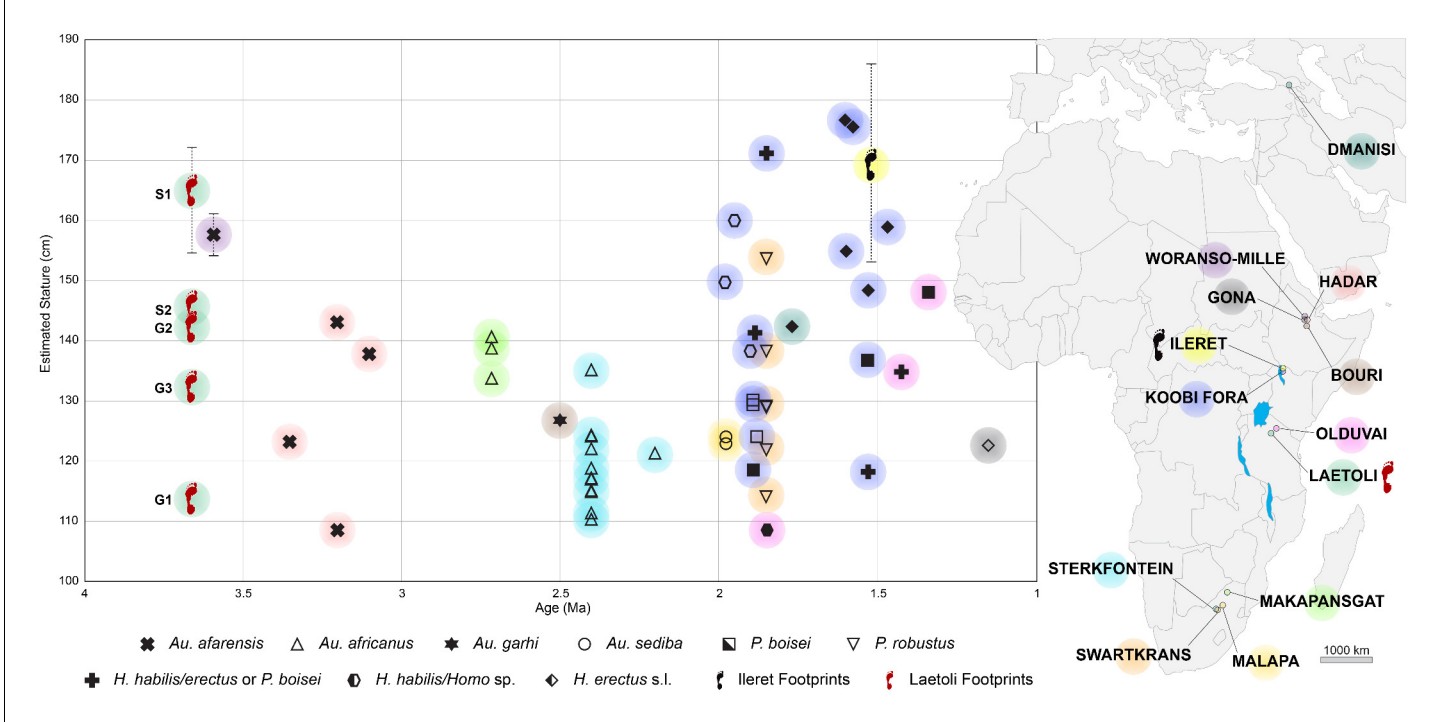

**Figure 12.** Estimates of predicted stature of fossil hominin individuals by species over time for the interval 4–1 Ma. Solid symbols (or crosses in bold) refer to stature estimates based on actual femur length; open symbols refer to stature estimates based on estimated femur length, in turn based on femur head diameter. For Laetoli and Ileret, stature estimates are based on footprint length (see Materials and methods). For Laetoli, Ileret and Woranso-Mille, the average value and range of predicted stature are shown. Colours are associated to the geographical location of each fossil/footprint site on the map. See *Supplementary file 5* for details.

same time, these data contrast with the hypothesis of a temporal trend of body-size increase among *Au. afarensis* between the more ancient Laetoli and the more recent Hadar fossil samples (*Lockwood et al., 2000*).

The impressive record of bipedal tracks from Laetoli Locality 8 (Site G and the new Site S) may open a window on the behaviour of a group of remote human ancestors, envisaging a scenario in which at least five individuals (G1, G2, G3, S1 and S2) were walking in the same time frame, in the same direction and at a similar moderate speed. This aspect must be evaluated in association with the pronounced body-size variation within the sample, which implies marked differences between age ranges and a considerable degree of sexual dimorphism in *Au. afarensis*. Significant implications about the social structure of this stem hominin species derive from these physical and behavioural characteristics, suggesting that reproductive strategies and social structure among at least some of the early bipedal hominins were closer to a gorilla-like model than to chimpanzees or modern humans.

Finally, the discovery reported here opens up the intriguing possibility that additional hominin trails may also occur in the area between Site G and Site S.

## Materials and methods

### Geology

Extended geological observations were carried out in the Laetoli area, mostly in the nearby historical Localities 6 and 7 (*Leakey, 1987b*), in order to compare the sequences exposed there with the new Site S sequence and to assess its stratigraphic position. Unfortunately, correlation with the stratigraphic sequence of Site G (Locality 8) is impossible because this historical site is completely covered by protection features and cannot be used for direct comparison.

In Site S, field observation and detailed sequence descriptions were carried out on excavation profiles following the standard formalized by *Catt (1990)*.

Basic observations on grain size, shape and mineralogy were carried out in the field using a 10x magnification hand lens. Higher-detail analyses were carried out in the laboratory, using a standard Leica stereomicroscope.

## Excavation and footprint imaging

The survey of the new tracks at Site S in September 2015 was focused on obtaining 3D models for documentation and morphometric analysis. The survey method is the *Structure from Motion* technique, an image-based process supported by in situ topographic measurements. This technique was chosen because of its technical advantages (relatively short time of data acquisition and processing; light and handy equipment; reduced costs) and excellent results in terms of resolution.

The equipment used in the fieldwork is a DSLR camera with 15.3 (4853 × 3198) megapixels and two different lenses: EF 24 mm f/2.8 for general shots of the excavations and EF 50 mm f/1.4 USM for details of the tracks. When necessary, the camera was mounted on a 4 m-long telescopic rod. A measuring tape and a water level were used for the measurement of the control points (i.e., circular targets with 35 mm diameter). Considering the small size of the surfaces to be detected, this measuring technique provided very high accuracy results.

## Fieldwork

The Ngorongoro Conservation Area Authority (NCAA), in whose jurisdiction the site is, provided the permit for the fieldwork as per letter with Ref. No. NCAA/D/157/Vol. IV of June 5, 2015.

Hominin and non-hominin tracks were recognised in four test-pits at Site S, namely L8, M9, TP2 and M10. The original 2 × 2 m square shape of L8 – the first test-pit where bipedal tracks were discovered – was modified during the study in order to follow the trail, and consequently took the complex shape in *Figure 2* (southern side: 2 m; western oblique side: 4 m). M9 was excavated some 14 m to the SSE of L8 and kept the planned size of 2 × 2 m. Following the interpolated alignment of the bipedal trackway, a third smaller test-pit, TP2 (1 × 1.2 m) (*Figure 6*) was excavated at some 8 m to the SSE of M9. Finally, a fourth test-pit, M9 (2 × 3 m) was excavated about 15 m to the east of M9 (*Figure 2*).

After the excavation, the 52 targets of the control point system were immediately positioned: 14 in L8, 10 in M9, 14 in TP2 and 14 in M10. Each test-pit was entirely surveyed at lower resolution and then detailed 3D models of some inner portions (single prints or groups of close prints) were acquired (*Figures 4–6*). We positioned four perimeter targets on the ground at the corners of each test-pit, and four inner targets around each sub-area surveyed in detail. The following list shows the target IDs in relation to the four test-pits and selected areas (AF: animal footprints):

- L8. Perimeter control points: A-B-C-D; footprint L8/S1-1: target 1–2-3–4; footprint L8/S1-2: target 3–4-5–6; footprint L8/S1-3: target 5–6-7–8; footprint L8/S1-4: target 7–8-9–10.
- M9. Perimeter control points: E-F-G-H; footprint M9/S1-2: target 21–22-23–24; footprint M9/S1-3: target 23–24-25–26.
- TP2. Perimeter control points: I-J-K-L; footprint TP2/S2-1: target 27–28-29–30; footprint TP2/S1-1: target 31–32-33–34; footprint TP2/S1-2: target 33–34-35–36.
- M10. Perimeter control points: M-N-O-P; AF1: target 11–12-13–14; AF2: target 13–15-19–20; AF3: target 15–16-17–18.

In order to optimize the timing of the fieldwork, we decided not to model in detail some of the hominin tracks, i.e. L8/S1-5 (visible only in its posterior portion), L8/S1-6 (virtually invisible due to the poor state of preservation of the Footprint Tuff), L8/S1-7 (damaged and excessively deep due to the lowering of the tuff cropping out on the scarp of the terrace), M9/S1-1 and M9/S1-4 (both filled by compact matrix).

In the second step, the perimeter target positions were measured. We placed two rods equipped with a spherical level on successive pairs of targets and we marked points at the same height on the rods for each pair by using the water level device. The vertical distance between these points and the targets, as well as their mutual distance, were recorded. Repeating this process for all pairs of targets, the relative plan position and the height of the control points were determined respectively by trilateration and by levelling.

A preliminary accuracy check was carried out during fieldwork, by using trilateration graphic rules in plan and by the method of successive levelling for heights. By assigning a z-coordinate to the first control point, all subsequent coordinates were derived from addition and subtraction of heights between two successive points. The check was performed by computing the algebraic sum of all height differences, and by verifying that the obtained value was close to zero. Finally, the error obtained in each test-pit was distributed to every z-coordinate of the points, in order to reduce it (*Supplementary file 1*).

The photographic survey was carried out by three shooting modes: (i) using the camera with the 24 mm lens, mounted on a telescopic rod at 4 m above the test-pits, in order to record each test-pit, as well as the spatial connection between test-pits; (ii) using the camera freehand with the 24 mm lens, in order to acquire additional shots of each test-pit; and (iii) using the camera close to the ground with the 50 mm lens, in order to acquire detailed sub-areas. More than 2,000 photos were taken, for a total of about 50 GB.

## Data processing

Data processing started by checking measurements in plan and height. This step is preliminary to the definition of the control point coordinates. The trilateration method was used to obtain x,y coordinates of the control points in plan. For each test-pit, six measurements were taken at the same height: the length of the four sides of the perimeter and the length of the two diagonals. Redundant measurements were used to compute the errors. In addition to a preliminary graphical control by CAD software (Autodesk AutoCAD), the automatic calculation software MicroSurvey STAR*NET was used to adjust, by least squares technique, a new set of x,y coordinates and heights of the control points (*Supplementary file 2*). The report provided by the software shows that the residues of adjustments never exceeded 10 mm (*Supplementary file 2*), which is a fully acceptable figure considering the size of the test-pits.

Once the adjusted x,y,z coordinate of all the control points (*Supplementary file 3*) were computed, we used them to scale and locate in the 3D space the 3D models built by the *Structure from Motion* technique (see below).

The pictures were first calibrated to reduce lens geometric distortion, and tone adjustment was applied in order to homogenize them and to reduce the effects of different lighting conditions during shooting. Subsequently, the software Agisoft Photoscan was used to generate 3D spatial data starting from the pictures, through the following phases: (i) alignment of the images; (ii) creation of the *dense point cloud*; (iii) transformation of the dense point cloud into a surface (*mesh*); (iv) application of the texture to the mesh (*Supplementary file 4*). A series of orthophotos (with and without textures) were extracted from the 3D models (*Figure 2—figure supplements 1*, *2* and *3* and *Figure 11—figure supplement 1*). A check on dense point cloud density was also carried out by Cloud-Compare, software for 3D point cloud and triangular mesh processing (*Figure 2—figure supplements 1*, *2* and *3* and *Figure 11—figure supplement 1*).

## Digital survey of the cast of the G1 and G2/G3 trails

At the end of the September 2015 field season, we also surveyed a first-generation fiberglass cast of the southern portion of the Site G trackway (about 4.7 m in length) (*Figure 11*) kept at the Leakey Camp at Olduvai Gorge. The cast includes the following tracks in the direction of walking: G1–39, 38, 37, 36, 35, 34, 33, 27, 26, 25 on the western side and G2/G3–31, 30, 29, 28, 27, 26, 25, 24, 20, 19 and 18 on the eastern side. Data acquisition and processing (*Supplementary file 4*) were performed following the workflow described above for the Site S test-pits. We positioned four perimeter control points and 11 inner targets. The latter were used to model in detail six selected tracks (G2/G3–29, G1–35, G1–34, G2/3–26, G2/3–25 and G2/3–18, listed in the direction of walking) (*Figure 11—figure supplement 1*).

## Morphometric analysis
### Morphometric data acquisition

The 3D data obtained by the above-explained procedures were also used in the morphometric analysis of the hominin tracks by Golden Software Surfer software. This contouring and surface modelling software transforms x,y,z data into maps (*Figures 4–6* and *11*). The x,y,z-format files were

imported into the software and transformed into grid files. The software uses randomly spaced x,y,z data to create regularly spaced grids composed of nodes with x,y,z coordinates. The *triangulation with linear interpolation* gridding method was applied, because it works best with data that are evenly distributed over the grid area. This method uses data points to create a network of triangles without edge intersections and computes new values along the edges. It is fast and does not extrapolate beyond the z-value of the data range; in addition, it assigns blanking values to grid nodes located outside the data limits. The grid spacing was set at 1 mm.

The following morphometric measures were taken on the contour maps:

- footprint length – maximum distance between the anterior tip of the hallux and the posterior tip of the heel;
- footprint max width – width across the distal metatarsal region;
- footprint heel width;
- angle of gait – angle between the midline of the trackway and the longitudinal axis of the foot;
- step length – distance between the posterior tip of the heel in two successive tracks;
- stride length – distance between the posterior tip of the heel in two successive tracks on the same side.

All of the above measurements were also taken manually both on the original tracks during the September 2015 field season, and on 1:1 scale sketches of the test-pits, hand-drawn on transparent plastic sheets. Morphometric values in *Table 2* are averaged from the results provided by the three methods described above in order to reduce errors. A synthesis of data extracted from *Table 2* is reported in *Table 3*. The foot index is defined as the percentage ratio between the max width and length of footprints.

## Morphometric data of the G1 and G2/G3 trails

Seventy human-like tracks arranged in two parallel trails (39 prints in G1 and 31 in G2/G3) are reported at Laetoli Site G (*Leakey, 1981*). Unfortunately, the whole set of morphometric data for the unearthed tracks was never published; only average values obtained from a selected number of tracks were provided. In the case of G2/G3, data are incomplete, largely because the prints of G3 are superimposed onto those of G2, so that it is difficult to collect the measurements (*Tuttle, 1987*). According to *Leakey (1981)*, only two (unspecified) prints of G2 are measurable. Morphometric data describing the Site G bipedal trails are summarized in *Table 3*, where they are also compared to the equivalent measurements taken on S1 and S2. Footprint length and maximum width for G1 and G3 are from *Tuttle (1987)* (average values obtained from nine and eight prints, respectively). Similar values are reported by *Leakey (1981)*, and slightly higher length values were recently published (*Bennett et al., 2016*) based on digital analysis of footprints casts (G1: 193 mm, N = 11; G3: 228 mm, N = 5). The length of G2 footprints (225 mm) is averaged from the two values of 210 and 240 mm reported for the only two measurable prints of G2 (*Leakey, 1981*). Similarly, the footprint max width of G2 (117 mm) is taken from *Leakey (1981)* (unknown sample size for this average). The average step and stride lengths for G1 and G3 are from *Tuttle (1987)*, whereas those for G2 are from *Robbins (1987)*.

## Stature, body mass and speed estimates

We used footprint size to estimate the stature of the Laetoli track-makers by means of different approaches. The easiest method follows *Tuttle (1987)* and consists of estimating the stature starting from the footprint length considering the ratio between foot length and stature in modern humans. Given that the foot length in *H. sapiens* is generally about 14% to 16% of stature (*Tuttle [1987]*, and references therein), we computed two estimates for the Laetoli hominins assuming that their feet were, respectively, 14% and 16% of their body height (*Tables 2–3*). This method, however, is not fully reliable because it is based on the body proportions of modern humans, and because it does not take into account that the footprint length does not accurately reflect the foot length. For this last reason, we also estimated stature using the method of *Dingwall et al. (2013)*, who published some equations based on regressions of stature by footprint length in modern Daasanach people (from the Lake Turkana area, Kenya). In particular, given the probable low walking speed of the Laetoli hominins (see below), we used the 'walk only' equation (Standard Error of Estimate, SEE = 5.4) (*Dingwall et al., 2013*). The obtained results (*Tables 2–3*) fall within the range of statures estimated

with the first method (except for G1 and G3, for which slightly taller statures were calculated). Finally, to assess how the results were influenced by considering modern human data, we also computed some estimates using the foot:stature ratio known for *Au. afarensis* (*Dingwall et al., 2013*). This ratio is 0.155–0.162 (*Dingwall et al., 2013*), so we obtained stature estimates (*Tables 2–3*) predictably close to or slightly lower than the lower limit of the estimates given by the *Tuttle (1987)* method.

Similarly, we estimated the body mass of the Laetoli track-makers using the 'walk only' regression equation that relates footprint area (i.e., footprint length x max. width) to body mass (SEE = 3.7) (*Dingwall et al., 2013*). For S2 only, we used the relationship between the footprint length and body mass (SEE = 3.8) (*Dingwall et al., 2013*) because of the enlarged morphology of TP2/S2-1. As for the stature, we re-calculated the mass using the known ratio between foot length and body mass in *Au. afarensis* (0.543–0.632) (*Dingwall et al. [2013]*, and references therein). The latter method resulted in estimates significantly lower than those computed by the aforementioned regression equation based on modern human data (*Tables 2* and *3*).

For both of the described methods, mean estimates of stature and body mass for S1 were computed by averaging the estimates obtained from individual tracks (*Tables 2* and *3*). The average footprint length values were considered more reliable than minimum values (which from a theoretical point of view could be regarded as more representative of the foot length) for the following reasons.

1. Previous studies demonstrated that footprint length can overestimate (*White and Suwa, 1987*) or underestimate (*Dingwall et al., 2013*) the actual foot length. Consequently, the average footprint length can be considered to be the most reliable parameter for the estimation of body dimensions (*White, 1980*; *Tuttle, 1987*; *Tuttle et al., 1990*; *Dingwall et al., 2013*; *Avanzini et al., 2008*; *Bennett et al., 2009*; *Roberts, 2009*).
2. In the specific case of the S1 trackway, the lengths of the three smaller tracks (*Table 2*) are probably underestimated: in L8/S1-1 (length: 250 mm) the anterior edge is poorly preserved and M9/S1-1 and M9/S1-4 (length: 245 mm) are still filled with sediment (see Introduction).

It must be pointed out that the stature and body-mass estimates for S2 must be considered with caution because they are based on a single preserved footprint. The same goes for G2, given the very low number of tracks for which the length can be measured (*Leakey, 1981*).

We also drew some inferences about the walking speed (*Table 3*), which is closely related to the stride length: in two individuals of the same body size, the one walking faster shows longer stride length. Nevertheless, the body proportions (i.e., stature, *h*) of the track-maker must be considered, because they influence the stride length (*L*) and consequently the velocity (*v*). We followed the power law computed by *Alexander (1976)*:

$$v = 0.25g^{0.5}L^{1.67}h^{-1.17} \tag{1}$$

where *g* is the gravitational acceleration (9.81 m/s$^2$). *Equation (1)* is widely used to estimate walking speed in humans and other animals (*Bennett and Morse [2014]*, and references therein).

Speed was further estimated following the method of *Dingwall et al. (2013)*. We used the regression equation that relates the speed to the ratio between stride length and average footprint length for each trail, obtaining values (*Table 3*) about twice those calculated with the equation (1). The transitional speed from walk to run is around 2.2 m/s (*Dingwall et al., 2013*). As the speed of the Laetoli track-makers is significantly lower than 2.2 m/s, we used the 'walk only' regression equation (*Dingwall et al., 2013*) for our speed estimates.

After computing the walking speed of S1 and G1–G3 with the aforementioned two methods, we obtained the relative speed (i.e., walking speed/estimated stature) (*Table 3*), which is a good indicator with which to compare the gait of different individuals regardless of their body proportions.

## Stature estimate comparisons

*Figure 12* was designed in order to compare graphically the stature estimates of the Laetoli individuals with those obtained for other hominin specimens. With the exception of the other footprint locality taken into account, Ileret in Kenya (*Bennett et al., 2009*; *Dingwall et al., 2013*), all other stature data are based on skeletal elements, namely femurs.

Early hominin stature reconstructions are notoriously difficult to assess: the limited number of intact long bones available in the fossil record often requires reconstruction of the long bone length from fragmentary remains, before different methods can be used to estimate the stature; the eventual results can differ according to the method employed. Thus, in an attempt to provide a synthetic picture of stature among australopithecines and early *Homo,* and to ensure that the results are comparable, we relied on a limited number of different datasets. Data are presented in *Supplementary file 5*.

For the geological age of the considered specimens and for their taxonomic attributions, we followed *Grabowski et al. (2015)*, unless otherwise indicated.

Two kinds of femur lengths were used for stature reconstruction: (i) the femur lengths of intact bones or femur length estimates based on reconstructions of incomplete bones; (ii) femur length estimates based on femur head diameters (FHD), according to the method given in *McHenry (1991)*. Morphometric data about complete or reconstructed femurs derive from *McHenry (1991)*, unless otherwise indicated (mostly fossils discovered after 1991). FHD values are from *Grabowski et al. (2015)*.

The two different equations cited in *McHenry (1991)* and in *Jungers et al. (2016)* were employed in stature reconstructions. As put into evidence in *Supplementary file 5*, the results are largely equivalent, with minor differences not relevant for the purpose of this analysis. Consequently, we used stature estimates obtained using the equation published by *Jungers et al. (2016)* to compile *Figure 12*.

## Access to material

Three-dimensional research-quality data are available from the MorphoSource digital repository (http://morphosource.org) without restrictions.

## Acknowledgements

This research is supported by the Italian School of Palaeoanthropology (University of Perugia; www.paleoantropologia.it), under the auspices of the Italian Ministry of Foreign Affairs and International Cooperation (*Italian archaeological, anthropological and ethnological missions abroad*) and the Italian Embassy in Dar es Salaam, Tanzania. The authors are grateful to the University of Dar es Salaam; the museum project consultant P Rich and the Ngorongoro Conservation Area Authority, without whom this discovery would have never been made; DM Kamamba, Director of Antiquities, Ministry of Natural Resources and Tourism; the University of Dar es Salaam for financial support; E Kazimoto for preliminary geological analysis; R Rettori for the organisation of the field season; S Grassi and A Grassi for 3D data processing and logistical support, R Pellizzon for photographs; P Blasi, M Lombardi, G Peter, L Quattrini, B Zamagni, A Songita and his assistants for the field work. R Blumenschine, M Haeusler, O Kullmer, J Njau, Y Rak, B Wood and R Wunderlich provided useful comments on an earlier version of the paper.

## Additional information

### Funding

| Funder | Author |
|---|---|
| Dipartimento di Fisica e Geologia, Università di Perugia | Marco Cherin<br>Angelo Barili<br>Giovanni Boschian<br>Dawid A Iurino<br>Sofia Menconero<br>Giorgio Manzi |
| Dipartimento di Biologia Ambientale, Sapienza Università di Roma | Giorgio Manzi |

The funders had no role in study design, data collection and interpretation, or the decision to submit the work for publication.

## Author contributions
FTM, EBI, Conception and design, Acquisition of data, Drafting or revising the article; MC, GM, Conception and design, Acquisition of data, Analysis and interpretation of data, Drafting or revising the article; AB, SM, Acquisition of data, Analysis and interpretation of data; GB, Acquisition of data, Analysis and interpretation of data, Drafting or revising the article; DAI, Conception and design, Acquisition of data, Analysis and interpretation of data; JM-C, Conception and design, Analysis and interpretation of data, Drafting or revising the article

## Author ORCIDs
Marco Cherin, ⓘ http://orcid.org/0000-0003-4291-4372
Sofia Menconero, ⓘ http://orcid.org/0000-0003-2136-5837

## Additional files

### Supplementary files
• Supplementary file 1. Footprint imaging, measurement report 1.

• Supplementary file 2. Footprint imaging, measurement report 2.

• Supplementary file 3. Footprint imaging, measurement report 3.

• Supplementary file 4. Footprint imaging, measurement report 4.

• Supplementary file 5. Individual fossil ages, localities and estimated statures used to build *Figure 12*.

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
