## [Decision Letter]

Thank you for submitting your article "New footprints from Laetoli (Tanzania) provide evidence for marked body size variation in early hominins" for consideration by *eLife*. Your article has been favorably evaluated by Ian Baldwin (Senior Editor) and three reviewers, one of whom, George H Perry (Reviewer #1), is a member of our Board of Reviewing Editors. The following individuals involved in review of your submission have agreed to reveal their identity: William Jungers (Reviewer #2) and Matthew Bennett (Reviewer #3).

The reviewers have discussed the reviews with one another and the Reviewing Editor has drafted this decision to help you prepare a revised submission.

Summary:

The reviewers agree that this paper provides both the description of a major discovery of new footprints of early bipedal hominins from Laetoli, Tanzania at 3.6 MYA and a valuable analytical result concerning early hominin body size variation. Laetoli is already famous for its Site G fossil footprints of (presumably) *Australopithecus afarensis* individuals. The new Site S footprints reported here represent very important additions to the Pliocene record of hominin behavior and morphology. A dozen new footprints from the S1 (N=11) and S2 (N=1) trails are sufficiently complete to estimate the body sizes of their makers (again, presumably *A. afarensis*) as well as approximate walking speeds. The new stature estimate for one of the individuals greatly exceeds those previously reconstructed from fossilized skeletal material or footprint data for any *A. afarensis* individual, consistent with substantial body size variation and likely sexual dimorphism within a single species of sexually dimorphic australopithecines.

Essential revisions:

The reviewers raise a several concerns that must be adequately addressed before the paper can be accepted. These points include the need to focus on the most appropriate body size estimation methods, to provide appropriate analysis and visualization of the comparisons of body size estimates among the different australopithecine individuals, and to revise the presentation of the manuscript to better reach a general, non-specialist audience.

1) The authors provide stature and mass estimates from the footprints using both modern human and australopithecine reference samples. The use of the modern human reference samples for stature estimates are inappropriate because they assume body proportions akin to modern humans, which are not possessed by australopithecines. Unfortunately, the australopith-based estimates are based on one individual, the tiny iconic female "Lucy" (A.L.288-1), and foot length in this individual is itself estimated. The authors should be more circumspect in reporting their results, acknowledging that the stature estimates from modern humans are likely exaggerations, and focus their interpretations on the more appropriate (but still tenuous) australopithecine-based predictions – still with the caveat about the limitations of the data from which the predictor is derived. Regardless, these footprints do expand the upper limit of size in (presumably) *A. afarensis*, and offer strong support for arguments favoring strong sexual size dimorphism in this extinct species.

2) The manuscript should provide additional data and analysis on the footprints and on the comparisons to previous australopithecine body size estimates. Table 3 should provide the range and standard deviation for print length, width, and index in addition to the average for S1, G1 and G3 trails. An ANOVA or other statistical test for significant differences in these parameters among the 3 trails should also be provided.

3) While there are 14 figures in the current version of the manuscript that contain images of the footprints themselves (see Essential Revisions comment #5 below), there are currently no figures depicting the key analytical result of the paper; that is, the large stature estimate of the S1 individual compared to estimates available for other australopithecines. This needs to be remedied. One option could be to depict stature estimates for all available australopithecines on the Y axis with MYA on the X axis, using different symbols for estimates derived from skeletal elements vs. footprints (and including the error estimates on the plot where available). Another part of the figure could illustrate the geographical location of each fossil/ footprint, with notation making it possible to reference each datapoint between the two panels.

4) The manuscript presentation requires rework, especially for general audience suitability. In the Introduction and Discussion, for non-specialist readers the significance of this result for interpretations of hominin behavior and evolutionary biology needs to be much more clear. There is not a need to list every theory; rather, focus on one or two of the main/ strongest implications (and the logic behind them, based on previous work) for early hominin behavior/ evolution based on the results of this study, laying out the case very clearly in the Discussion (after presenting possibilities of both sides in the Introduction). In the Results, the main results need more attention. For example, it is insufficient to simply (and briefly) refer the reader to the Table 2 and Table 3 for the main results of the study. Also, some brief explanation for how body mass and stature are calculated should be provided in this section (Essential Revisions comment #1 above is also relevant here).

5) Many of the 14 figures are gorgeous, and the images and descriptions of the find are an important part of the paper. However, the key findings and results would be more apparent to the reader if there were some streamlining of the figures. Some of the existing figures could be integrated as panels into others (e.g., there are many figures with images of the S1 track and different footprints that could potentially be combined) or using the *eLife* feature for embedded supplementary figures.

6) Please ensure that the scans of the footprints and images (at the full resolution available and used for analyses in the study) are made publicly available through an appropriate data repository, and report the database accession details in the revised manuscript.

---

## [Author Response]

*[…] Essential revisions:*

*The reviewers raise a several concerns that must be adequately addressed before the paper can be accepted. These points include the need to focus on the most appropriate body size estimation methods, to provide appropriate analysis and visualization of the comparisons of body size estimates among the different australopithecine individuals, and to revise the presentation of the manuscript to better reach a general, non-specialist audience.*

*1) The authors provide stature and mass estimates from the footprints using both modern human and australopithecine reference samples. The use of the modern human reference samples for stature estimates are inappropriate because they assume body proportions akin to modern humans, which are not possessed by australopithecines. Unfortunately, the australopith-based estimates are based on one individual, the tiny iconic female "Lucy" (A.L.288-1), and foot length in this individual is itself estimated. The authors should be more circumspect in reporting their results, acknowledging that the stature estimates from modern humans are likely exaggerations, and focus their interpretations on the more appropriate (but still tenuous) australopithecine-based predictions – still with the caveat about the limitations of the data from which the predictor is derived. Regardless, these footprints do expand the upper limit of size in (presumably) A. afarensis, and offer strong support for arguments favoring strong sexual size dimorphism in this extinct species.*

We thank the reviewer for this. A new sentence was added in the manuscript to highlight that we mainly focused our interpretations on the australopithecine-based prediction method. For this reason, the estimated stature of S1 cited in the discussion was changed from about 170 to 165 cm (subsection “Implications of the new Laetoli footprints”). Finally, we formatted in bold type the australopithecine-based estimated statures and body masses in Table 3.

*2) The manuscript should provide additional data and analysis on the footprints and on the comparisons to previous australopithecine body size estimates. Table 3 should provide the range and standard deviation for print length, width, and index in addition to the average for S1, G1 and G3 trails. An ANOVA or other statistical test for significant differences in these parameters among the 3 trails should also be provided.*

Table 3 was amended as requested by the reviewers. Unfortunately, we could not perform any statistical test for significant differences because in the literature the individual measurements of the G footprints were *never published*, but only average values and, at most, ranges (see Leakey, 1981; Tuttle, 1987).

*3) While there are 14 figures in the current version of the manuscript that contain images of the footprints themselves (see Essential Revisions comment #5 below), there are currently no figures depicting the key analytical result of the paper; that is, the large stature estimate of the S1 individual compared to estimates available for other australopithecines. This needs to be remedied. One option could be to depict stature estimates for all available australopithecines on the Y axis with MYA on the X axis, using different symbols for estimates derived from skeletal elements vs. footprints (and including the error estimates on the plot where available). Another part of the figure could illustrate the geographical location of each fossil/ footprint, with notation making it possible to reference each datapoint between the two panels.*

We are grateful to the reviewers for highlighting this. We built a new figure (Figure 12) in which the estimated stature of hominin individuals is plotted against time (Ma). We decided to consider not only australopithecines, but also some early *Homo* individuals, in order to emphasise that the estimated stature of S1 can be comparable to that of more derived taxa, such as *Homo erectus sensu lato*. We used different symbols for estimations derived from footprints vs. skeletal elements, namely femurs. In the latter case, we also indicated graphically if the estimates are based on actual vs. estimated femur length. Each point in the graphic is also linked to the corresponding palaeontological locality through a colour code. Data used to build the figure are in the new [Supplementary-material SD5-data]. We commented on the figure in the Discussion (subsection “Implications of the new Laetoli footprints”) and wrote a dedicated section in the Materials and methods (subsection “Stature estimate comparisons”).

*4) The manuscript presentation requires rework, especially for general audience suitability. In the Introduction and Discussion, for non-specialist readers the significance of this result for interpretations of hominin behavior and evolutionary biology needs to be much more clear. There is not a need to list every theory; rather, focus on one or two of the main/ strongest implications (and the logic behind them, based on previous work) for early hominin behavior/ evolution based on the results of this study, laying out the case very clearly in the Discussion (after presenting possibilities of both sides in the Introduction). In the Results, the main results need more attention. For example, it is insufficient to simply (and briefly) refer the reader to the Table 2 and Table 3 for the main results of the study. Also, some brief explanation for how body mass and stature are calculated should be provided in this section (Essential Revisions comment #1 above is also relevant here).*

We think that this is definitely a good suggestion. We extensively modified the Abstract and Introduction, trying to provide a comprehensive background on the subject of intraspecific morphological variation in hominins and on how it can influence biological and behavioural aspects. In addition, in the Introduction, we discussed the basis for the scientific debate on the variation in *Au. afarensis*, in order to enable the reader to better understand the value of our conclusions.

Introductive notes about Laetoli were moved to a new section (The site: a brief overview), in which some additional information on this important site were added for non-specialist readers.

In the Results, we added a sentence (subsection “Speed, stature and body mass estimates”) to stress the main results of our stature and body mass estimation. The sentence in the aforementioned subsection briefly explains how stature and body mass were calculated.

The Discussion section was modified and expanded to emphasise the importance of our results in the context of the debate on the variation in *Au. afarensis*, also adding some recent references on the subject (e.g., Maslin et al., 2015; Jungers et al., 2016). Moreover, we started from the body size estimated data to add some inferences on biological aspects, such as sexual dimorphism and social structure.

*5) Many of the 14 figures are gorgeous, and the images and descriptions of the find are an important part of the paper. However, the key findings and results would be more apparent to the reader if there were some streamlining of the figures. Some of the existing figures could be integrated as panels into others (e.g., there are many figures with images of the S1 track and different footprints that could potentially be combined) or using the eLife feature for embedded supplementary figures.*

Previous Figure 12–13 and 14 are now supplementary to Figure 11 and Figure 2, respectively.

*6) Please ensure that the scans of the footprints and images (at the full resolution available and used for analyses in the study) are made publicly available through an appropriate data repository, and report the database accession details in the revised manuscript.*

The 3D models of the test-pits with hominin tracks will be uploaded to MorphoSource (http://morphosource.org) as soon as the paper is accepted.